# Investigating how ReLU-networks encode symmetries

**Georg Bökman    Fredrik Kahl**
Chalmers University of Technology
{bokman, fredrik.kahl}@chalmers.se

## Abstract

Many data symmetries can be described in terms of group equivariance and the most common way of encoding group equivariances in neural networks is by building linear layers that are group equivariant. In this work we investigate whether equivariance of a network implies that all layers are equivariant. On the theoretical side we find cases where equivariance implies layerwise equivariance, but also demonstrate that this is not the case generally. Nevertheless, we conjecture that CNNs that are trained to be equivariant will exhibit layerwise equivariance and explain how this conjecture is a weaker version of the recent permutation conjecture by Entezari et al. [2022]. We perform quantitative experiments with VGG-nets on CIFAR10 and qualitative experiments with ResNets on ImageNet to illustrate and support our theoretical findings. These experiments are not only of interest for understanding how group equivariance is encoded in ReLU-networks, but they also give a new perspective on Entezari et al.'s permutation conjecture as we find that it is typically easier to merge a network with a group-transformed version of itself than merging two different networks.

.

## 1   Introduction

Understanding the inner workings of deep neural networks is a key problem in machine learning and it has been investigated from many different perspectives, ranging from studying the loss landscape and its connection to generalization properties to understanding the learned representations of the networks. Such an understanding may lead to improved optimization techniques, better inductive biases of the network architecture and to more explainable and predictable results. In this paper, we focus on ReLU-networks—networks with activation function $\texttt{ReLU}(x) := \max(0, x)$—and how they encode and learn data symmetries via equivariance.

Equivariances can be built into a neural network by design. The most classical example is of course the CNN where translational symmetry is obtained via convolutional layers. Stacking such equivariant layers in combination with pointwise $\texttt{ReLU}$ activations results in an equivariant network. We will pursue a different research path and instead start with a neural network which has been trained to be equivariant and ask how the equivariance is encoded in the network. This approach in itself is not new and has been, for instance, experimentally explored in [24] where computational methods for quantifying layerwise equivariance are developed. We will shed new light on the problem by deriving new theoretical results when network equivariance implies layerwise equivariance. We will also give counterexamples for when this is not the case. Another insight is obtained via the recent conjecture by Entezari et al. [13] which states that networks with the same architecture trained on the same data are often close to each other in weight space modulo permutation symmetries of the network weights. Our new conjecture 3.2 which we derive from the conjecture of Entezari et al. states that most SGD CNN solutions will be close to group equivariant CNNs (GCNNs).

37th Conference on Neural Information Processing Systems (NeurIPS 2023).

**Net A**                              **Net B**

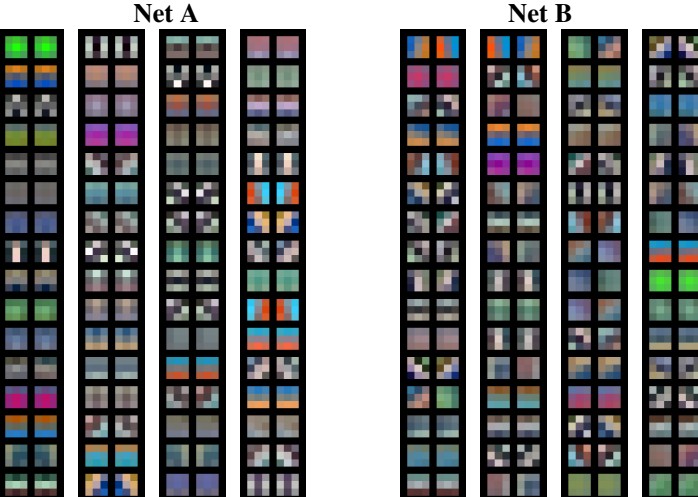

Figure 1: Illustration of how a VGG11 encodes the horizontal flipping symmetry. The 64 filters in the first convolutional layer of two VGG11-nets trained on CIFAR10 are shown, where each filter is next to a filter in the same net which after horizontally flipping the filter results in a similar convolution output. The order of the filters in the right columns is a permutation of the original order in the left columns. This permutation is obtained with the method in Figure 2, i.e., the columns here correspond to **1.** and **4.** there. **Net A** is trained with an invariance loss to output the same logits for horizontally flipped images. It has learnt very close to a GCNN structure where each filter in the first layer is either horizontally symmetric or has a mirrored twin. **Net B** is trained with horizontal flipping data augmentation. This net is quite close to a GCNN structure, but the mirrored filters are less close to each other than in Net A. More details are given in Section 4.

**1.  2.  3.  4.**

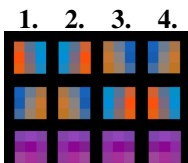

Figure 2: Permutation aligning horizontally flipped filters.
**1.** The three original filters in a convolutional layer.
**2.** The three filters flipped horizontally.
**3.** Permutation of the flipped filters to align them with the originals. This permutation is found using activation matching following [25].
**4.** Flipping the filters back to their original form for illustration.

Our theoretical results and new conjecture are also validated with experiments. We focus exclusively on the horizontal flipping symmetry in image classification, for the following three reasons: 1. Horizontal flipping is ubiquitously assumed to not change the class of an image and horizontal flipping data augmentation is in practice always used. 2. The group of horizontal flips ($S_2$) is small and has simple representation theory. 3. As we study `ReLU` as activation function, if a network is layerwise equivariant, then the representations acting on the feature spaces must be permutation representations as we shall prove later (cf. Section 2.1).

The experiments show that networks trained on CIFAR10 and ImageNet with horizontal flipping data augmentation are close to GCNNs and in particular when we train networks with an invariance loss, they become very close to GCNNs. This is illustrated in Figure 1 and also discussed in Section 4. As further support of our main conjecture, we find that the interpolation barrier is lower for merging a ResNet50 with a flipped version of itself than the barrier for merging two separately trained ResNet50's. In summary, our main contribution is a new conjecture on how ReLU-networks encode symmetries which is supported by both theory and experiments.

## 1.1   Related work

**Group equivariant neural networks.** The most common approach to encoding group symmetries into neural networks is by designing the network so that each layer is group equivariant [41, 14]. This has been done for many groups, e.g., the permutation group [43, 27] and the 2D or 3D rotation group [39, 34, 16, 4, 37]. Another possibility is to use symmetry regularization during training [32]. Group

equivariant nets have many possible applications, for instance estimating molecular properties [31] or image feature matching [3]. For this paper, the most relevant group equivariant nets are so-called GCNNs acting on images, which are equivariant to Euclidean transformations [10, 42, 2, 40, 38]. In the experiments in Section 4 we will investigate how close to GCNNs ordinary CNNs are when they are trained to be equivariant—either by using an invariance loss or by using data augmentation.

**Measuring layerwise equivariance.** [24] measure layerwise equivariance by fitting linear group representations in intermediate feature spaces of networks. Our approach is similar but we restrict ourselves to permutation representations and consider fitting group representations in all layers simultaneously rather than looking at a single layer at a time. We explain in Section 2.1 why it is enough to search for permutation representations, given that we have a network with `ReLU` activation functions. [5] measure layerwise equivariance by counting how many filters in each layer have group-transformed copies of themselves in the same layer. They find that often more layerwise equivariance means better performance. Our approach explicitly looks for permutations of the filters in each layer that align the filters with group-transformed versions of themselves. We are thus able to capture how close the whole net is to being a GCNN. [29] find evidence for layerwise equivariance using a qualitative approach of visualizing filters in each layer and finding which look like group-transformed versions of each other. [20] measure local layerwise equivariance, i.e., layerwise robustness to small group transformations, by computing derivatives of the output of a network w.r.t. group transformations of the input. They interestingly find that transformers can be more translation equivariant than CNNs, due to aliasing effects in CNN downsampling layers.

**Networks modulo permutation symmetries.** [25] demonstrated that two CNNs with the same architecture trained on the same data often learn similar features. They did this by permuting the filters in one network to align with the filters in another network. [13] conjectured that it should be possible to permute the weights of one network to put it in the same loss-basin as the other network and that the networks after permutation should be linearly mode connected. I.e., it should be possible to average ("merge") the weights of the two networks to obtain a new network with close to the same performance. This conjecture has recently gained empirical support in particular through [1], where several good methods for finding permutations were proposed and [22] where the performance of a merged network was improved by resetting batch norm statistics and batch statistics of individual neurons to alleviate what the authors call variance collapse. In Section 3 we give a new version of [13]'s conjecture by conjecturing that CNNs trained on group invariant data should be close to GCNNs.

## 1.2 Limitations

While we are able to show several relevant theoretical results, we have not been able to give a conclusive answer in terms of necessary and sufficient conditions to when equivariance of a network implies layerwise equivariance or that the network can be rewritten to be layerwise equivariant. An answer to this question would be valuable from a theoretical point of view.

In the experiments, we limit ourselves to looking at a single symmetry of images – horizontal flipping. As in most prior work on finding weight space symmetries in trained nets, we only search for permutations and not scaled permutations which would also be compatible with the `ReLU`-nonlinearity. Our experimental results depend on a method of finding permutations between networks that is not perfect [25]. Future improvements to permutation finding methods may increase the level of certainty which we can have about Conjectures 3.1 and 3.2.

## 2 Layerwise equivariance

We will assume that the reader has some familiarity with group theory and here only briefly review a couple of important concepts. Given a group $G$, a representation of $G$ is a group homomorphism $\rho : G \to \mathrm{GL}(V)$ from $G$ to the general linear group of some vector space $V$. E.g., if $V = \mathbb{R}^m$, then $\mathrm{GL}(V)$ consists of all invertible $m \times m$-matrices and $\rho$ assigns a matrix to each group element, so that the group multiplication of $G$ is encoded as matrix multiplication in $\rho(G) \subset \mathrm{GL}(V)$.

A function $f : V_0 \to V_1$ is called *equivariant* with respect to a group $G$ with representations $\rho_0$ on $V_0$ and $\rho_1$ on $V_1$ if $f(\rho_0(g)x) = \rho_1(g)f(x)$ for all $g \in G$ and $x \in V_0$. An important representation that exists for any group on any vector space is the trivial representation where $\rho(g) = I$ for all $g$. If

a function $f$ is equivariant w.r.t. $\rho_0$ and $\rho_1$ as above, and $\rho_1$ is the trivial representation, we call $f$ *invariant*. A *permutation representation* is a representation $\rho$ for which $\rho(g)$ is a permutation matrix for all $g \in G$. We will also have use for the concept of a group invariant data distribution. We say that a distribution $\mu$ on a vector space $V$ is $G$-invariant w.r.t. a representation $\rho$ on $V$, if whenever $X$ is a random variable distributed according to $\mu$, then $\rho(g)X$ is also distributed according to $\mu$.

Let's for now[1] consider a neural network $f : \mathbb{R}^{m_0} \to \mathbb{R}^{m_L}$ as a composition of linear layers $W_j : \mathbb{R}^{m_{j-1}} \to \mathbb{R}^{m_j}$ and activation functions $\sigma$:

$$f(x) = W_L \sigma(W_{L-1}\sigma(\cdots W_2\sigma(W_1 x)\cdots)). \tag{1}$$

Assume that $f$ is equivariant with respect to a group $G$ with representations $\rho_0 : G \to \mathrm{GL}(\mathbb{R}^{m_0})$ on the input and $\rho_L : G \to \mathrm{GL}(\mathbb{R}^{m_L})$ on the output, i.e.,

$$f(\rho_0(g)x) = \rho_L(g)f(x), \quad \text{for all } x \in \mathbb{R}^{m_0}, g \in G. \tag{2}$$

A natural question to ask is what the equivariance of $f$ means for the layers it is composed of. Do they all have to be equivariant? For a layer $W_j$ to be $G$-equivariant we require that there is a representation $\rho_{j-1}$ on the input and a representation $\rho_j$ on the output of $W_j$ such that $\rho_j(g)W_j = W_j\rho_{j-1}(g)$ for all $g \in G$. Note again that the only representations that are specified for $f$ to be equivariant are $\rho_0$ and $\rho_L$, so that all the other $\rho_j : G \to \mathrm{GL}(\mathbb{R}^{m_j})$ can be arbitrarily chosen. If there exists a choice of $\rho_j$'s that makes each layer in $f$ equivariant (including the nonlinearities $\sigma$), we call $f$ *layerwise equivariant*. In general we could have that different representations act on the input and output of the nonlinearities $\sigma$, but as we explain in Section 2.1, this cannot be the case for the ReLU-nonlinearity, which will be our main focus. Hence we assume that the the same group representation acts on the input and output of $\sigma$ as above.

The following simple example shows that equivariance of $f$ does not imply layerwise equivariance. *Example* 2.1. Let $G = S_2 = \{i, h\}$ be the permutation group on two indices, where $i$ is the identity permutation and $h$ the transposition of two indices. Consider a two-layer network $f : \mathbb{R}^2 \to \mathbb{R}$,

$$f(x) = W_2\, \texttt{ReLU}(W_1 x),$$

where $W_1 = (1 \quad 0)$ and $W_2 = 0$. $f$ is invariant to permutation of the two coordinates of $x$ (indeed, $f$ is constant 0). Thus, if we select $\rho_0(h) = \left(\begin{smallmatrix} 0 & 1 \\ 1 & 0 \end{smallmatrix}\right)$ and $\rho_2(h) = 1$, then $f$ is equivariant (note that a representation of $S_2$ is specified by giving an involutory $\rho(h)$ as we always have $\rho(i) = I$). However, there is no choice of a representation $\rho_1$ that makes $W_1$ equivariant since that would require $\rho_1(h)W_1 = W_1\rho_0(h) = (0 \quad 1)$, which is impossible (note that $\rho_1(h)$ is a scalar). The reader will however notice that we can define a network $\tilde{f}$, with $\tilde{W}_1 = (0 \quad 0)$, $\tilde{W}_2 = 0$ for which we have $f(x) = \tilde{f}(x)$ and then $\tilde{f}$ is layerwise equivariant when choosing $\rho_1(h) = 1$.

In the example just given it was easy to, given an equivariant $f$, find an equivalent net $\tilde{f}$ which is layerwise equivariant. For very small 2-layer networks we can prove that this will always be the case, see Proposition D.1.

Example 2.1 might seem somewhat unnatural, but we note that the existence of "dead neurons" (also known as "dying ReLUs") with constant zero output is well known [26]. Hence, such degeneracies could come into play and make the search for linear representations in trained nets more difficult. We will however ignore this complication in the experiments in the present work.

From an intuitive point of view, equivariance of a neural network should mean that some sort of group action is present on the intermediate feature spaces as the network should not be able to "forget" about the equivariance in the middle of the net only to recover it at the end. In order to make this intuition more precise we switch to a more abstract formulation in Appendix D.1. The main takeaway will be that it is indeed possible to define group actions on modified forms of the intermediate feature spaces whenever the network is equivariant, but this will not be very practically useful as it changes the feature spaces from vector spaces to arbitrary sets, making it impossible to define linear layers to/from these feature spaces.

An interesting question that was posed by Elesedy and Zaidi [12], is whether non-layerwise equivariant nets can ever perform better at equivariant tasks than layerwise equivariant nets. In Appendix C, we give a positive answer by demonstrating that when the size of the net is low, equivariance can hurt

---

[1]The main results in this section are generalized to networks with affine layers in Appendix E.

performance. In particular we give the example of *cooccurence of equivariant features* in C.1. This is a scenario in image classification, where important features always occur in multiple orientations in every image. It is intuitive that in such a case, it suffices for the network to recognize a feature in a single orientation for it to be invariant *on the given data*, but a network recognizing features in only one orientation will not be layerwise equivariant.

## 2.1 From general representations to permutation representations

We will now first sidestep the issue of an equivariant network perhaps not being layerwise equivariant, by simply assuming that the network is layerwise equivariant with a group representation acting on every feature space (this will to some degree be empirically justified in Section 4). Then we will present results on 2-layer networks, where we can in fact show that layerwise equivariance is implied by equivariance.

The choice of activation function determines which representations are at all possible. In this section we lay out the details for the `ReLU`-nonlinearity. We will have use for the following lemma, which is essentially the well known property of `ReLU` being "positive homogeneous".

**Lemma 2.2.** *[Godfrey et al. [18, Lemma 3.1, Table 1]] Let $A$ and $B$ be invertible matrices such that* $\mathtt{ReLU}(Ax) = B\,\mathtt{ReLU}(x)$ *for all $x$. Then $A = B = PD$ where $P$ is a permutation matrix and $D$ is a diagonal matrix with positive entries on the diagonal.*

We provide an elementary proof in Appendix B.1. It immediately follows from Lemma 2.2 that if `ReLU` is $G$-equivariant with respect to representations $\rho_0$ and $\rho_1$ on the input and output respectively, then $\rho_0(g) = \rho_1(g) = P(g)D(g)$ for all $g \in G$. I.e., the representations acting on the intermediate feature spaces in a layerwise equivariant `ReLU`-network are scaled permutation representations. This holds also if we add bias terms to the layers in (1).

We note that Godfrey et al. [18] consider more nonlinearities than `ReLU`, and that their results on other nonlinearities could similarly be used to infer what input and output group representations are admissible for these other nonlinearities. Also, Wood and Shawe-Taylor [41] derive in large generality what nonlinearities commute with which finite group representations, which is very related to our discussion. However, to apply the results from [41] we would have to first prove that the representations acting on the inputs and outputs of the nonlinearity are the same.

For two-layer networks with invertible weight matrices we can show that equivariance implies layerwise equivariance with a scaled permutation representation acting on the feature space on which `ReLU` is applied. The reader should note that the invertibility assumption is strong and rules out cases such as Example 2.1.

**Proposition 2.3.** *Consider the case of a two-layer network $f : \mathbb{R}^m \to \mathbb{R}^m$,*

$$f(x) = W_2\,\mathtt{ReLU}(W_1 x),$$

*where `ReLU` is applied point-wise. Assume that the matrices $W_1 \in \mathbb{R}^{m \times m}$, $W_2 \in \mathbb{R}^{m \times m}$ are non-singular. Then $f$ is $G$-equivariant with $\rho_j : G \to \mathrm{GL}(\mathbb{R}^m)$ for $j = 0, 2$ on the input and the output respectively if and only if*

$$\rho_0(g) = W_1^{-1} P(g)D(g)W_1 \quad and \quad \rho_2(g) = W_2 P(g)D(g)W_2^{-1},$$

*where $P(g)$ is a permutation and $D(g)$ a diagonal matrix with positive entries. Furthermore, the network $f$ is layerwise equivariant with $\rho_1(g) = P(g)D(g)$.*

*Proof.* Invertibility of $W_1$ and $W_2$ means that $f$ being equivariant w.r.t. $\rho_0$, $\rho_2$ is equivalent to `ReLU` being equivariant w.r.t. $\rho_1(g) = W_1\rho_0(g)W_1^{-1}$ and $\tilde\rho_1(g) = W_2^{-1}\rho_2(g)W_2$. The discussion after Lemma 2.2 now shows that $\rho_1(g) = \tilde\rho_1(g) = P(g)D(g)$ and the proposition follows. □

The set of group representations for which a two-layer ReLU-network can be $G$-equivariant is hence quite restricted. It is only representations that are similar (or conjugate) to scaled permutations that are feasible. In the appendix, we also discuss the case of two-layer networks that are $G$-invariant and show that they have to be layerwise equivariant with permutation representations in Proposition B.1.

## 2.2 Permutation representations in CNNs—group convolutional neural networks

As explained in Section 2.1, for a `ReLU`-network to be layerwise equivariant, the representations must be scaled permutation representations. We will now review how such representations can be used to encode the horizontal flipping symmetry in CNNs. This is a special case of group equivariant convolutional networks—GCNNs—which were introduced by [10]. An extensive reference is [38]. The reader familiar with [10, 38] can skip this section, here we will try to lay out in a condensed manner what the theory of horizontal flipping equivariant GCNNs looks like.

Note first that horizontally flipping an image corresponds to a certain permutation of the pixels. Let's denote the action of horizontally flipping by $\tau$. This is a permutation representation of the abstract group $S_2$. If a convolutional layer $\Psi$ only contains horizontally symmetric filters, then it is immediate that $\Psi(\tau(x)) = \tau(\Psi(x))$. It is however possible to construct more general equivariant layers.

Let's assume that the representations acting on the input and output of $\Psi$ split into the spatial permutation $\tau$ and a permutation $P$ of the channels. $P_0$ on the input and $P_1$ on the output. $P_0$ and $P_1$ need to have order maximum 2 to define representations of $S_2$. One can show, using the so-called kernel constraint [39, 38], what form the convolution kernel $\psi$ of $\Psi$ needs to have to be equivariant. The kernel constraint says that the convolution kernel $\psi \in \mathbb{R}^{c_1 \times c_0 \times k \times k}$ needs to satisfy

$$\tau(\psi) = P_1 \psi P_0^T \ (= P_1 \psi P_0) \,, \tag{3}$$

where $\tau$ acts on the spatial $k \times k$ part of $\psi$, $P_1$ on the $c_1$ output channels and $P_0$ on the $c_0$ input channels. The last equality holds since $P_0$ is of order 2. In short, permuting the channels of $\psi$ with $P_0$ and $P_1$ should be the same as horizontally flipping all filters. We see this in action in Figure 1, even for layers that are not explicitly constrained to satisfy (3). An intuitive explanation of why (3) should hold for equivariant layers is that it guarantees that the same information will be captured by $\psi$ on input $x$ and flipped input $\tau(x)$, since all horizontally flipped filters in $\tau(\psi)$ do exist in $\psi$.

Channels that are fixed under a channel permutation are called invariant and channels that are permuted are called regular, as they are part of the regular representation of $S_2$. We point out three special cases—if $P_0 = P_1 = I$, then $\psi$ has to be horizontally symmetric. If $P_0 = I$ and $P_1$ has no diagonal entries, then we get a lifting convolution and if $P_0$ and $P_1$ both have no diagonal entries then we get a regular group convolution [10]. In the following when referring to a *regular* GCNN, we mean the "most generally equivariant" case where $P$ has no diagonal entries.

## 3 The permutation conjecture by Entezari et al. and its connection to GCNNs

Neural networks contain permutation symmetries in their weights, meaning that given, e.g., a neural network of the form (1), with pointwise applied nonlinearity $\sigma$, we can arbitrarily permute the inputs and outputs of each layer

$$W_1 \mapsto P_1 W_1, \quad W_j \mapsto P_j W_j P_{j-1}^T, \quad W_L \mapsto W_L P_{L-1}^T,$$

and obtain a functionally equivalent net. The reader should note the similarity to the kernel constraint (3). It was conjectured by [13] that given two nets of the same type trained on the same data, it should be possible to permute the weights of one net to put it in the same loss-basin as the other net. Recently, this conjecture has gained quite strong empirical support [1, 22].

When two nets are in the same loss-basin, they exhibit close to linear mode connectivity, meaning that the loss/accuracy barrier on the linear interpolation between the weights of the two nets will be close to zero. Let the weights of two nets be given by $\theta_1$ and $\theta_2$ respectively. In this paper we define the barrier of a performance metric $\zeta$ on the linear interpolation between the two nets by

$$\frac{\frac{1}{2}(\zeta(\theta_1) + \zeta(\theta_2)) - \zeta\left(\frac{1}{2}(\theta_1 + \theta_2)\right)}{\frac{1}{2}(\zeta(\theta_1) + \zeta(\theta_2))} . \tag{4}$$

Here $\zeta$ will most commonly be the test accuracy. Previous works [15, 13, 1, 22] have defined the barrier in slightly different ways. In particular they have not included the denominator which makes it difficult to compare scores for models with varying performance. We will refer to (4) without the denominator as the absolute barrier. Furthermore we evaluate the barrier only at a single interpolation point—halfway between $\theta_1$ and $\theta_2$—as compared to earlier work taking the maximum of barrier values when interpolating between $\theta_1$ and $\theta_2$. We justify this by the fact that the largest barrier value

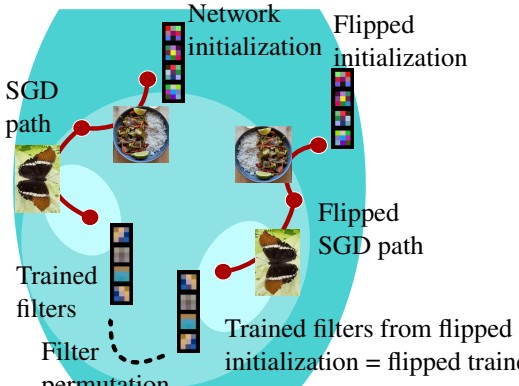

Figure 3: For every CNN that is trained with horizontal flipping data augmentation, there is a corresponding equally likely CNN that was initialized with filters horizontally flipped and trained on horizontally flipped images. This CNN is the same as the original but with flipped filters, also after training. According to the permutation conjecture [13], there should be a permutation of the channels of the flipped CNN that aligns it close to the original CNN. This implies that the CNN is close to a GCNN. Lighter blue means a spot in the parameter landscape with higher accuracy.

is practically almost always halfway between $\theta_1$ and $\theta_2$ (in fact [13] also only evaluate the halfway point in their experiments). The permutation conjecture can now be informally stated as follows.

**Conjecture 3.1** (Entezari et al. [13, Sec. 3.2]). *Most SGD solutions belong to a set $\mathcal{S}$ whose elements can be permuted in such a way that there is no barrier on the linear interpolation between any two permuted elements in $\mathcal{S}$.*

Importantly, we note that when applied to CNNs, the conjecture should be interpreted in the sense that we only permute the channel dimensions of the convolution weights, not the spatial dimensions. We will now explain why applying Conjecture 3.1 to CNNs that are trained on invariant data distributions leads to a conjecture stating that most SGD solutions are close to being GCNNs. For simplicity, we discuss the case where $G = $ horizontal flips of the input image, but the argument works equally well for vertical flips or 90 degree rotations. The argument is summarised in Figure 3.

We will consider an image classification task and the common scenario where images do not change class when they are horizontally flipped, and where a flipped version of an image is equally likely as the original (in practice this is very frequently enforced using data augmentation). Let $\tau$ be the horizontal flipping function on images. We can also apply $\tau$ to feature maps in the CNN, whereby we mean flipping the spatial horizontal dimension of the feature maps.

Initialize one CNN $\Phi$ and copy the initialization to a second CNN $\hat{\Phi}$, but flip all filters of $\hat{\Phi}$ horizontally. If we let $x_0, x_1, x_2, \ldots$ be the samples drawn during SGD training of $\Phi$, then an equally likely drawing of samples for SGD training of $\hat{\Phi}$ is $\tau(x_0), \tau(x_1), \tau(x_2), \ldots$. After training $\Phi$ using SGD and $\hat{\Phi}$ using the horizontally flipped version of the same SGD path, $\hat{\Phi}$ will still be a copy of $\Phi$ where all filters are flipped horizontally.[2] This means according to Conjecture 3.1 that $\Phi$ can likely be converted close to $\hat{\Phi}$ by only permuting the channels in the network.

Let's consider the $j$'th convolution kernel $\psi \in \mathbb{R}^{c_j \times c_{j-1} \times k \times k}$ of $\Phi$. There should exist permutations $P_j$ and $P_{j-1}$ such that $\tau(\psi) \approx P_j \psi P_{j-1}^T$, where the permutations act on the channels of $\psi$ and $\tau$ on the spatial $k \times k$ part. But if we assume that the $P_j$'s are of order 2 so that they define representations of the horizontal flipping group, this is precisely the kernel constraint (3), which makes the CNN close to a GCNN! It seems intuitive that the $P_j$'s are of order 2, i.e. that if a certain channel should be permuted to another to make the filters as close to each other as possible then it should hold the other way around as well. However, degeneracies such as multiple filters being close to equal can in practice hinder this intuition. Nevertheless, we conclude with the following conjecture which in some sense is a weaker version of Conjecture 3.1, as we deduced it from that one.

**Conjecture 3.2.** *Most SGD CNN-solutions on image data with a distribution that is invariant to horizontal flips of the images will be close to GCNNs.*

---

[2]This is not strictly true of networks that contain stride-2 convolutions or pooling layers with padding, as they typically start at the left edge of the image and if the image has an even width won't reach to the right edge of the image, meaning that the operation is nonequivalent to performing it on a horizontally flipped input. Still it will be approximately equivalent. We will discuss the implications of this defect in Section A.2.1.

Table 1: Types of VGG11 nets considered. All except "w/o aug" are trained with horizontal flipping augmentation.

| Name | Description |
| --- | --- |
| CNN | Ordinary VGG11 trained using cross-entropy loss (C.-E.). 9.23M parameters. |
| CNN w/o aug | Ordinary VGG11 trained without horizontal flipping augmentation. |
| CNN + inv-loss | Ordinary VGG11 trained using C.-E. and invariance loss (inv-loss). |
| CNN + late inv-loss | Ordinary VGG11 trained using C.-E. and inv-loss after 20% of the epochs. |
| GCNN | A regular horizontal flipping GCNN trained using C.-E. 4.61M parameters. |
| PGCNN | A partial horizontal flipping GCNN trained using C.-E. The first two conv-layers are $G$-conv-layers and the rest are ordinary conv-layers. 9.19M parameters. |
| PGCNN + late inv-loss | The PGCNN trained using C.-E. and inv-loss after 20% of the epochs. |

**A measure for closeness to being a GCNN.** To measure how close a CNN $\Phi$ is to being a GCNN we can calculate the barrier, as defined in (4), of the linear interpolation between $\Phi$ and a permutation of the flipped version of $\Phi$. We call this barrier for the test accuracy the *GCNN barrier*.

## 4 Experiments

The aim of the experimental section is to investigate two related questions.

(Q1) If a CNN is trained to be invariant to horizontal flips of input images, is it a GCNN? This question is related to the theoretical investigation of layerwise equivariance in Section 2.

(Q2) If a CNN is trained on a horizontal flipping-invariant data distribution, will it be close to a GCNN? This is Conjecture 3.2.

To answer these two questions we evaluate the GCNN barrier for CNNs trained on CIFAR10 and ImageNet. We look at CNNs trained with horizontal flipping data augmentation to answer (Q2) and CNNs trained with an invariance loss on the logits to answer (Q1). In fact, for all training recipes horizontal flipping data augmentation is used. The invariance loss applied during training of a CNN $\Phi$ is given by $\|\Phi(x) - \Phi(\tau(x))\|$, where $\tau$ horizontally flips $x$. It is added to the standard cross-entropy classification loss. To evaluate the invariance of a CNN $\Phi$ we will calculate the relative invariance error $\|\Phi(x) - \Phi(\tau(x))\|/(0.5\|\Phi(x)\| + 0.5\|\Phi(\tau(x))\|)$, averaging over a subset of the training data. Another way to obtain invariance to horizontal flips is to use some sort of self-supervised learning approach. We will investigate self-supervised learning of ResNets in Section 4.2.

To align networks we will use activation matching [25, 1]. In activation matching, the similarity between channels in feature maps of the same layer in two different networks is measured over the training data and a permutation that aligns the channels as well as possible between the networks according to this similarity is found. Furthermore, we will use the REPAIR method by [22], which consists of—after averaging the weights of two networks—reweighting each channel in the obtained network to have the average batch statistics of the original networks. This reweighting can be merged into the weights of the network so that the original network structure is preserved. REPAIR is a method to compensate for the fact that when two filters are not perfect copies of each other, the variance of the output of their average will be lower than the variance of the output of the original filters. We will also report results without REPAIR.

Experimental details can be found in Appendix A. We provide code for merging networks with their flipped selfs at `https://github.com/georg-bn/layerwise-equivariance`.

### 4.1 VGG11 on CIFAR10

We train VGG11 nets [33] on CIFAR10 [23]. We will consider a couple of different versions of trained VGG11 nets, they are listed in Table 1.

We train 24 VGG11 nets for each model type and discard crashed runs and degenerate runs[3] to obtain 18-24 good quality nets of each model type. In Figure 1 we visualize the filters of the first layer in two

---

[3]By degenerate run we mean a net with accuracy 10%. There was one GCNN and one CNN + late inv-loss for which this happened.

Table 2: Statistics for VGG11 nets trained on CIFAR10.

| Name | Accuracy | Invariance Error | GCNN Barrier |
|------|----------|------------------|--------------|
| CNN | $0.901 \pm 2.1 \cdot 10^{-3}$ | $0.282 \pm 1.8 \cdot 10^{-2}$ | $4.00 \cdot 10^{-2} \pm 4.9 \cdot 10^{-3}$ |
| CNN w/o aug | $0.879 \pm 1.8 \cdot 10^{-3}$ | $0.410 \pm 4.0 \cdot 10^{-2}$ | $4.98 \cdot 10^{-2} \pm 6.1 \cdot 10^{-3}$ |
| CNN + inv-loss | $0.892 \pm 2.3 \cdot 10^{-3}$ | $0.0628 \pm 6.7 \cdot 10^{-3}$ | $1.90 \cdot 10^{-3} \pm 8.9 \cdot 10^{-4}$ |
| CNN + late inv-loss | $0.902 \pm 2.8 \cdot 10^{-3}$ | $0.126 \pm 1.6 \cdot 10^{-2}$ | $1.33 \cdot 10^{-2} \pm 2.8 \cdot 10^{-3}$ |
| GCNN | $0.899 \pm 2.5 \cdot 10^{-3}$ | $1.34 \cdot 10^{-6} \pm 1.5 \cdot 10^{-7}$ | $8.92 \cdot 10^{-4} \pm 1.6 \cdot 10^{-3}$ |
| PGCNN | $0.902 \pm 3.7 \cdot 10^{-3}$ | $0.274 \pm 2.0 \cdot 10^{-2}$ | $3.09 \cdot 10^{-2} \pm 7.8 \cdot 10^{-3}$ |
| PGCNN + late inv-loss | $0.915 \pm 2.3 \cdot 10^{-3}$ | $0.124 \pm 1.5 \cdot 10^{-2}$ | $7.56 \cdot 10^{-3} \pm 1.8 \cdot 10^{-3}$ |

Table 3: Results for ResNet50's trained using various methods on ImageNet. The model with an asterisk is trained by us. The last five are self-supervised methods. Flip-Accuracy is the accuracy of a net with all conv-filters horizontally flipped. Halfway Accuracy is the accuracy of a net that has merged the weights of the original net and the net of the net with flipped filters—after permuting the second net to align it to the first. For the accuracy and barrier values w/o REPAIR we still reset batch norm statistics after merging by running over a subset of the training data.

| Training Method | Invariance Error | Accuracy | Flip-Accuracy | Halfway Accuracy, no REPAIR | Halfway Accuracy | GCNN Barrier, no REPAIR | GCNN Barrier |
|-----------------|------------------|----------|---------------|-----------------------------|------------------|-------------------------|--------------|
| Torchvision new | 0.284 | 0.803 | 0.803 | 0.731 | 0.759 | 0.0893 | 0.0545 |
| Torchvision old | 0.228 | 0.761 | 0.746 | 0.718 | 0.725 | 0.0467 | 0.0384 |
| *Torchvision old + inv-loss | 0.0695 | 0.754 | 0.745 | 0.745 | 0.745 | 0.00636 | 0.0054 |
| BYOL | 0.292 | 0.704 | 0.712 | 0.573 | 0.624 | 0.19 | 0.118 |
| DINO | 0.169 | 0.753 | 0.744 | 0.611 | 0.624 | 0.184 | 0.166 |
| Moco-v3 | 0.16 | 0.746 | 0.735 | 0.64 | 0.681 | 0.136 | 0.0805 |
| Simsiam | 0.174 | 0.683 | 0.667 | 0.505 | 0.593 | 0.252 | 0.121 |

VGG11 nets. More such visualizations are presented in Appendix A. We summarise the statistics of the trained nets in Table 2. The most interesting findings are that the models with low invariance error also have low GCNN barrier, indicating that invariant models are layerwise equivariant. Note that the invariance error and GCNN barrier should both be zero for a GCNN. In order to have something to compare the numbers in Table 2 to, we provide barrier levels for merging two different nets in Table 4 in the appendix. Of note is that merging a model with a flipped version of itself is consistently easier than merging two separate models. In Figure 4 in the appendix, we show the distribution of permutation order for channels in different layers. The matching method sometimes fails to correctly match GCNN channels (for which we know ground truth matches), but overall it does a good job. In general, the unconstrained nets seem to learn a mixture between invariant and regular GCNN channels. Preliminary attempts with training GCNN-VGG11s with such a mixture of channels did however not outperform regular GCNNs.

## 4.2   ResNet50 on ImageNet

Next we look at the GCNN barrier for ResNet50 [21] trained on ImageNet [11]. We consider a couple of different training recipes. First of all two supervised methods—the latest Torchvision recipe [35] and the old Torchvision recipe [36] which is computationally cheaper. Second, four self-supervised methods: BYOL [19], DINO [7], Moco-v3 [9] and Simsiam [8]. The results are summarised in Table 3. There are a couple of very interesting takeaways. First, the GCNN barrier for the supervised methods is unexpectedly low. When merging two separate ResNet50's trained on ImageNet, [22] report an absolute barrier of 20 percentage points, whereas we here see barriers of 4.4 percentage points for the new recipe and 2.9 percentage points for the old recipe. This indicates that it easier to merge a net with a flipped version of itself than with a different net. However, we also observe that for the self-supervised methods the barrier is quite high (although still lower than 20 percentage points). This is curious since they are in some sense trained to be invariant to aggressive data augmentation—including horizontal flips. Finally we note that when training with an invariance loss, the GCNN barrier vanishes, meaning that the obtained invariant net is close to being a GCNN.

The reader may have noticed that the "Flip-Accuracy", i.e., accuracy of nets with flipped filters, is markedly different than the accuracy for most nets in Table 3. This is a defect stemming from using image size 224. We explain this and present a few results with image size 225 in Appendix A.2.1.

## 5    Conclusions and future work

We studied layerwise equivariance of ReLU-networks theoretically and experimentally. Theoretically we found both positive and negative results in Section 2, showing that layerwise equivariance is in some cases guaranteed given equivariance but in general not. In Section 3, we explained how Entezari et al.'s permutation conjecture 3.1 can be applied to a single CNN and the same CNN with horizontally flipped filters. From this we extrapolated Conjecture 3.2 stating that SGD CNN solutions are likely to be close to being GCNNs, also leading us to propose a new measure for how close a CNN is to being a GCNN—the GCNN barrier. In Section 4 we saw quite strong empirical evidence for the fact that a ReLU-network that has been trained to be equivariant will be layerwise equivariant. We also found that it is easier to merge a ResNet with a flipped version of itself, compared to merging it with another net. Thus, Conjecture 3.2 might be a worthwhile stepping stone for the community investigating Conjecture 3.1, in addition to being interesting in and of itself. If a negative GCNN barrier is achievable, it would imply that we can do "weight space test time data augmentation" analogously to how merging two separate nets can enable weight space ensembling.

**Acknowledgements**

This work was partially supported by the Wallenberg AI, Autonomous Systems and Software Program (WASP) funded by the Knut and Alice Wallenberg Foundation. The computations were enabled by resources provided by the National Academic Infrastructure for Supercomputing in Sweden (NAISS) at the Chalmers Centre for Computational Science and Engineering (C3SE) partially funded by the Swedish Research Council through grant agreement no. 2022-06725 and by the Berzelius resource provided by the Knut and Alice Wallenberg Foundation at the National Supercomputer Centre.

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

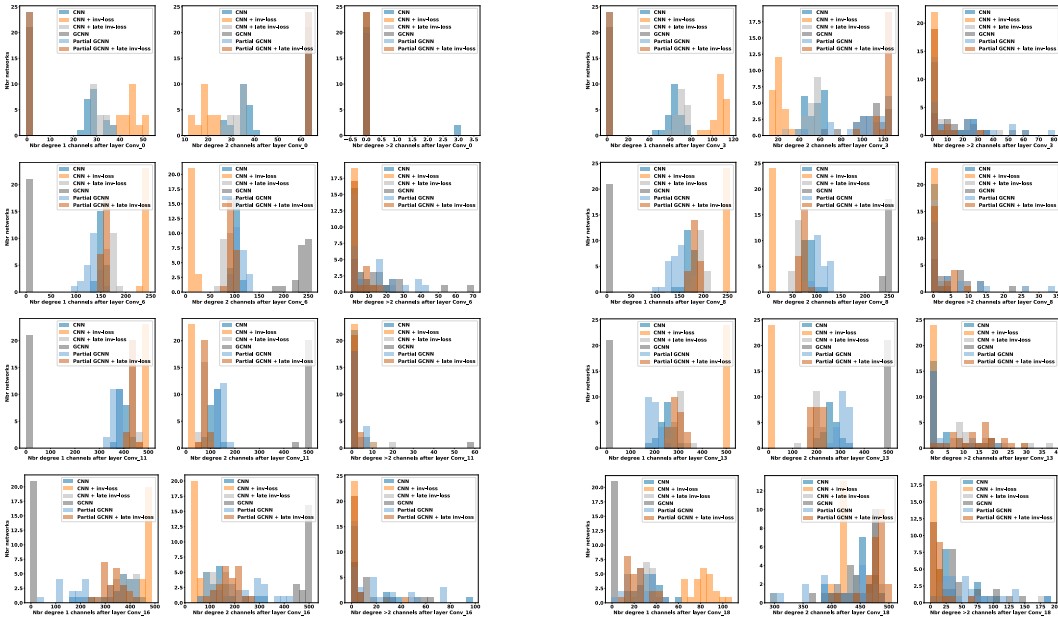

Figure 4: Number of channels of different permutation order that are found by activation matching in different VGG11 versions. In the ideal case, for GCNNs all channels should have order $\leq 2$. The matching method fails this, but not extremely badly. The main reason for failure that we have observed is that layers contain many zero-filters.

# A    Experimental details and more results

## A.1    VGG11 on CIFAR10

We present the distribution of various channels types in Figure 4. We present statistics for merging two different nets in Table 4.

In Figures 5, 6 we show the convolution kernels of the second convolution layers of the same nets as in Figure 1. Figures 7, 8 show the filters of a GCNN-VGG11, which has hardcoded equivariance in each layer.

There are some interesting takeaways from these filter visualizations. First of all, the number of zero-filters in Figure 6 – second layer of an ordinary VGG11 – and Figure 8 – second layer of a GCNN-VGG11 – is quite high and it is lower for the VGG11 trained using invariance loss – Figure 5. Also, the GCNN has more zero-filters than the ordinary net. It would be valuable to investigate why these zero-filters appear and if there is some training method that doesn't lead to such apparent under-utilization of the network parameters.

Second, naturally when enforcing maximally equivariant layers using a lifting convolution in the first layer (recall Section 2.2), the network is forced to learn duplicated horizontal filters – see Figure 7 e.g. the all green filter. This could explain the worse performance of GCNN:s relative to ordinary VGG11:s. The choice of representations to use in each layer of a GCNN remains a difficult problem, and as we mentioned in the main text, in our initial experiments it did not seem to work to set the representations similar to those observed in Figure 4. Further research on the optimal choice of representations would be very valuable. However, as discussed in Section C.1, it may be the case that no layerwise equivariant net outperforms nets which are not restricted to be equivariant in late layers.

### A.1.1    Training details

The nets were trained on NVIDIA T4 GPUs on a computing cluster. We train 4 nets in parallel on each GPU, each run taking between 1 and 2 hours, depending on net type. This yields less than $(24/4) \cdot 6 \cdot 2 = 72$ compute hours in total.

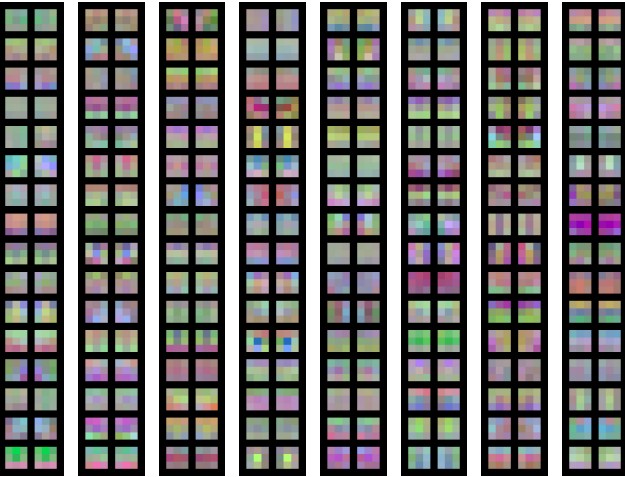

Figure 5: Illustration of how a VGG11 encodes the horizontal flipping symmetry. The 128 filters in the second convolutional layer of a VGG11-net trained on CIFAR10 are shown, where each filter is next to a filter in a permuted version of the horizontally flipped convolution kernel. Both the input and output channels of the flipped convolution kernel are permuted, as described in Section 3. The 64 input channels are projected by a random projection matrix to 3 dimensions for visualization as RGB. The net is trained with an invariance loss to output the same logits for horizontally flipped images. It has learnt very close to a GCNN structure where the horizontally flipped filters can be permuted close to the original.

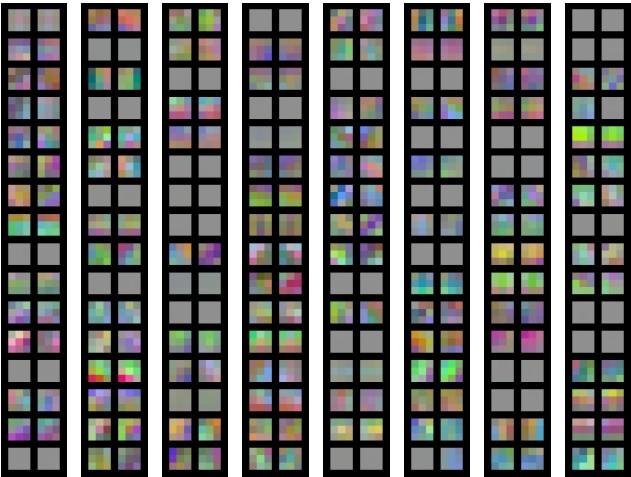

Figure 6: Illustration of how a VGG11 encodes the horizontal flipping symmetry. The 128 filters in the second convolutional layer of a VGG11-net trained on CIFAR10 are shown, where each filter is next to a filter in a permuted version of the horizontally flipped convolution kernel. Both the input and output channels of the flipped convolution kernel are permuted, as described in Section 3. The 64 input channels are projected by a random projection matrix to 3 dimensions for visualization as RGB. The net is trained with horizontal flipping data augmentation. This net is quite close to a GCNN structure, but the permuted filters are less close to each other than in Figure 5

Table 4: Statistics for merging VGG11 nets that where trained in different manners.

| Model 1 | Model 2 | Barrier |
|---|---|---|
| CNN | CNN | $5.08 \cdot 10^{-2} \pm 5.7 \cdot 10^{-3}$ |
| CNN | CNN + inv-loss | $4.87 \cdot 10^{-2} \pm 4.5 \cdot 10^{-3}$ |
| CNN | GCNN | $6.50 \cdot 10^{-2} \pm 8.9 \cdot 10^{-3}$ |
| CNN + inv-loss | CNN + inv-loss | $3.73 \cdot 10^{-2} \pm 4.1 \cdot 10^{-3}$ |
| CNN + inv-loss | GCNN | $7.53 \cdot 10^{-2} \pm 9.6 \cdot 10^{-3}$ |
| GCNN | GCNN | $6.59 \cdot 10^{-2} \pm 7.7 \cdot 10^{-3}$ |



Figure 7: Illustration of how a GCNN-VGG11 encodes the horizontal flipping symmetry. The 64 filters in the first convolutional layer of a GCNN-VGG11-net trained on CIFAR10 are shown, where each filter is next to a filter in a permuted version of the horizontally flipped convolution kernel.

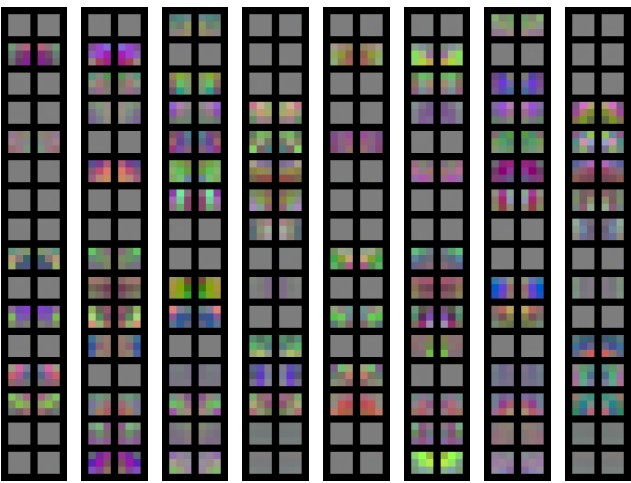

Figure 8: Illustration of how a GCNN-VGG11 encodes the horizontal flipping symmetry. The 128 filters in the second convolutional layer of a GCNN-VGG11-net trained on CIFAR10 are shown, where each filter is next to a filter in a permuted version of the horizontally flipped convolution kernel. Both the input and output channels of the flipped convolution kernel are permuted, as described in Section 3. The 64 input channels are projected by a random projection matrix to 3 dimensions for visualization as RGB.

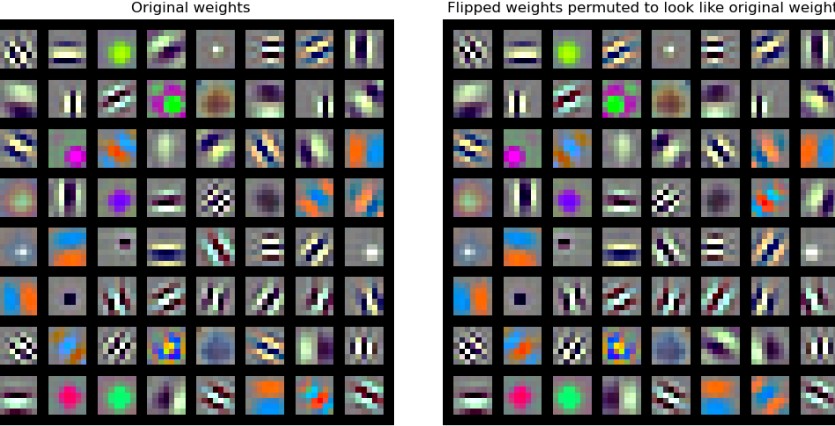

Figure 9: Left: Visualization of the filters in the first layer of a ResNet50. Right: The same filters but horizontally flipped and permuted to align with the original filters to the right.

## A.2   ResNet50 on ImageNet

We show the first layer filters of the torchvision new recipe ResNet50 in Figure 9.

### A.2.1   The peculiarities of image size 224

As mentioned in footnote 2, a convolution layer with stride 2 can not be equivariant to horizontal flipping if the input has even width. This problem has been discussed previously by [28, 30]. All the methods in Table 3 except for the new Torchvision recipe use image size 224 for training and all use it for testing. It so happens that 224 can be halved 5 times without remainder meaning that all feature spaces in a ResNet50 will have even width. As luck would have it, if we use image size 225, then all feature spaces have odd width instead. Therefore we report results for a few methods with image size 225 in Table 5, this table contains all numbers from Table 3 as well for convenience. We see that the simple change of image size lowers the invariance score and GCNN barrier. However it does not seem to influence the accuracy much, meaning that the networks seem to learn to compensate for the non-equivariance with image size 224.

### A.2.2   Training details

We train on NVIDIA A100 GPUs on a computing cluster. Each run was on either 4 or 8 parallel GPUs. For the supervised net, training took around between 9 and 13 hours depending on image size and whether invariance loss was applied. For the Simsiam training, the self-supervised pre-training took 23 hours and training the supervised linear head took 15 hours. In total this gives an upper bound of $13 \cdot 8 \cdot 3 + (15 + 23) \cdot 8 = 616$ GPU hours.

Table 5: Results for ResNet50's trained using various methods on ImageNet. The models with an asterisk are trained by us. The last five are self-supervised methods. Flip-Accuracy is the accuracy of a net with all conv-filters horizontally flipped. For the accuracy and barrier values w/o REPAIR we still reset batch norm statistics after merging by running over a subset of the training data.

| Training Method | Invariance Error | Accuracy | Flip-Accuracy | Halfway Accuracy w/o RE-PAIR | Halfway Accuracy | GCNN Barrier w/o RE-PAIR | GCNN Barrier |
|---|---|---|---|---|---|---|---|
| Torchvision new | 0.284 | 0.803 | 0.803 | 0.731 | 0.759 | 0.0893 | 0.0545 |
| Torchvision old | 0.228 | 0.761 | 0.746 | 0.718 | 0.725 | 0.0467 | 0.0384 |
| *Torchvision old + inv-loss | 0.0695 | 0.754 | 0.745 | 0.745 | 0.745 | 0.00636 | 0.0054 |
| *Torchvision old + img size 225 | 0.162 | 0.764 | 0.764 | 0.714 | 0.722 | 0.0657 | 0.0558 |
| *Torchvision old + inv-loss + img size 225 | 0.00235 | 0.744 | 0.745 | 0.744 | 0.744 | $9.4 \cdot 10^{-5}$ | $6.72 \cdot 10^{-5}$ |
| BYOL | 0.292 | 0.704 | 0.712 | 0.573 | 0.624 | 0.19 | 0.118 |
| DINO | 0.169 | 0.753 | 0.744 | 0.611 | 0.624 | 0.184 | 0.166 |
| Moco-v3 | 0.16 | 0.746 | 0.735 | 0.64 | 0.681 | 0.136 | 0.0805 |
| Simsiam | 0.174 | 0.683 | 0.667 | 0.505 | 0.593 | 0.252 | 0.121 |
| *Simsiam + img size 225 | 0.113 | 0.68 | 0.68 | 0.535 | 0.611 | 0.213 | 0.103 |

# B  Layerwise equivariance of ReLU-networks – proofs and further results

## B.1  Proof of Lemma 2.2

**Lemma 2.2.** *[Godfrey et al. [18, Lemma 3.1, Table 1]] Let $A$ and $B$ be invertible matrices such that $\mathrm{ReLU}(Ax) = B\,\mathrm{ReLU}(x)$ for all $x$. Then $A = B = PD$ where $P$ is a permutation matrix and $D$ is a diagonal matrix with positive entries on the diagonal.*

We will use the following notation. For a scalar $a$, we can uniquely write it as $a = a^+ - a^-$, where $a^+ \geq 0$ and $a^- \geq 0$, and either $a^+ = 0$ or $a^- = 0$. Similarly for matrices (elementwise), $A = A^+ - A^-$.

*Proof.* As the equation $\mathrm{ReLU}(Ax) = B\,\mathrm{ReLU}(x)$ should hold for all $x$, we can start by setting $x = -e_k$ where $e_k$ is the canonical basis. Then, as $\mathrm{ReLU}(-e_k) = 0$, we get that $\mathrm{ReLU}(-Ae_k) = A_k^- = 0$. Hence, $A^- = 0$ and consequently $A = A^+$. Inserting $x = e_k$ into the equation, we can conclude that $A = B = A^+$.

Now suppose that two elements on the same row are non-zero, say $A_{ik} > 0$ and $A_{ik'} > 0$ on row $i$ for some $k \neq k'$. Let's see what happens when we set $x = A_{ik}e_{k'} - A_{ik'}e_k$. On the left hand side, the $i$:th element simply becomes $\mathrm{ReLU}(A_{ik}A_{ik'} - A_{ik'}A_{ik}) = 0$ whereas on the right hand side we get $A\,\mathrm{ReLU}(A_{ik}e_{k'} - A_{ik'}e_k) = A\,\mathrm{ReLU}(A_{ik}e_{k'}) = A_{ik}A_{k'}$, that is, the $i$:th element is $A_{ik}A_{ik'} \neq 0$, which is a contradiction. Hence, we can conclude that each row has at most one positive element and since the matrices $A$ and $B$ are non-singular, they will have exactly one positive element in each row and we can conclude that they must be scaled permutation matrices.

□

## B.2  A result on layerwise equivariance given an invariant network

Next we are going to look at two-layer networks $f : \mathbb{R}^m \to \mathbb{R}$, which are invariant, that is, $G$-equivariant with $\rho_2(g) = I$. Now, if $f(x) = W_2\,\mathrm{ReLU}(W_1 x)$, one can without loss of generality assume that it is only the signs of the elements in $W_2$ that matter, and move the magnitudes into $W_1$ and write $f(x) = [\pm 1 \quad \dots \quad \pm 1]\,\mathrm{ReLU}(\tilde{W}_1 x)$. Also, one can sort the indices such that $W_2 = [1 \quad \dots \quad 1 \quad \text{-}1 \quad \dots \quad \text{-}1]$. This leads to a convenient characterization of invariant two-layer networks.

**Proposition B.1.** *Consider the case of a two-layer network* $f : \mathbb{R}^m \to \mathbb{R}$,

$$f(x) = W_2 \, \mathtt{ReLU}(W_1 x),$$

*where* $\mathtt{ReLU}$ *is applied point-wise. Assume that the matrix* $W_1 \in \mathbb{R}^{m \times m}$ *is non-singular and that* $W_2 \in \mathbb{R}^{m \times 1}$ *has the form* $[1 \;\; \ldots \;\; 1 \;\; \text{-}1 \;\; \ldots \;\; \text{-}1]$ *with* $m_+$ *positive elements and* $m_-$ *negative elements* $(m = m_+ + m_-)$. *Further, assume* $\rho_2(g) = I$ *for the output, that is, the trivial representation. Then,* $f$ *is* $G$-*equivariant with* $\rho_0 : G \to \mathrm{GL}(\mathbb{R}^m)$ *if and only if*

$$\rho_0(g) = W_1^{-1} \begin{bmatrix} P_+(g) & 0 \\ 0 & P_-(g) \end{bmatrix} W_1,$$

*where* $P_+(g)$ *is an* $m_+ \times m_+$ *permutation matrix and* $P_-(g)$ *an* $m_- \times m_-$ *permutation matrix.*
*Furthermore, the network* $f$ *is layerwise equivariant with* $\rho_1(g) = \begin{bmatrix} P_+(g) & 0 \\ 0 & P_-(g) \end{bmatrix}$.

*Proof.* The condition for invariance is that the following equation should hold for all $g \in G$ and all $x \in \mathbb{R}^m$,

$$W_2 \, \mathtt{ReLU}(W_1 x) = W_2 \, \mathtt{ReLU}(W_1 \rho_0(g) x).$$

Now, let $y = W_1 x$ and $A = W_1 \rho_0(g) W_1^{-1}$, we get that the following equivalent condition

$$W_2 \, \mathtt{ReLU}(y) = W_2 \, \mathtt{ReLU}(Ay)$$

should hold for all $A = A(g) = W_1 \rho_0(g) W_1^{-1}$. In particular, it should hold for $y = e_k - \lambda e_{k'}$ for $k \neq k'$ and $\lambda > 0$. For this choice, the left hand side is equal to $\pm 1$ and hence the right hand side should also be independent of $\lambda$. The right hand side has $m_+$ positive terms and $m_-$ negative terms,

$$\sum_{i=1}^{m_+} (A_{ik} - \lambda A_{ik'})^+ - \sum_{i=m_++1}^{m} (A_{ik} - \lambda A_{ik'})^+.$$

One possibility is that one of the positive terms cancels out a negative one. However, this would imply that $A$ would contain two identical rows, making $A$ singular, which is not feasible. Now, if $A_{ik'} < 0$, then for sufficiently large $\lambda$, it would follow that $(A_{ik} - \lambda A_{ik'})^+ = A_{ik} - \lambda A_{ik'}$ which makes the right hand side dependent on $\lambda$, a contradiction. Hence, all the elements of $A$ must be positive. Now, if both $A_{ik} >$ and $A_{ik'} > 0$, then for sufficiently small $\lambda$, we get $(A_{ik} - \lambda A_{ik'})^+ = A_{ik} - \lambda A_{ik'}$, again a contradiction. Therefore, at most one element can be positive on each row. By also checking $y = e_k$, one can draw the conclusion that $A$ must be a permutation matrix of the required form and the result follows. $\square$

## C  Answer to an open question by Elesedy and Zaidi

A question which was proposed by Elesedy and Zaidi [12, Sec. 7.4] is whether non-layerwise equivariant nets can ever be better at equivariant tasks than layerwise equivariant nets. We saw in Example 2.1 that they can be equally good, but not whether they can be better.

We formulate the question as follows.

(C.Q) Given a $G$-invariant data distribution $\mu$, and a ground truth equivariant function $s : \mathbb{R}^{m_0} \to \mathbb{R}^{m_L}$, can we ever have for neural networks of the form (1) that

$$\inf_{W_j \in \mathbb{R}^{m_{j-1} \times m_j}} \mathbb{E}_{x \sim \mu} \| f_{\{W_j\}}(x) - s(x) \| < \inf_{\substack{W_j \in \mathbb{R}^{m_{j-1} \times m_j}, \\ W_j \text{ equivariant}}} \mathbb{E}_{x \sim \mu} \| f_{\{W_j\}}(x) - s(x) \|?$$

Here we assume that the layer dimensions $m_j$ are fixed and equal on both hand sides. We write $f_{\{W_j\}}$ to make the parameterisation of $f$ in terms of the linear layers explicit.

We will now give a simple example showing that the answer to the question is yes.

*Example* C.1. Let $\mu$ be a distribution on $\mathbb{R}$ that is symmetric about the origin and let $s : \mathbb{R} \to \mathbb{R}$ be given by $s(x) = |x|$. $s$ is invariant to changing the sign of the input, i.e., equivariant to $S_2 = \{i, h\}$

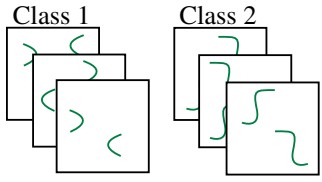

Figure 10: Illustration of what we term *cooccurence of equivariant features*. We imagine training a network to classify the two depicted classes. Both classes consist of images containing a shape and horizontally flipped copies of the same shape – in particular, every image contains the shape in both orientations. On this data, a network only recognising the existence of the shape $\subset$ (in this orientation) will be invariant to horizontal flips of the input. However, such a network is not invariant to horizontal flips of general data and can hence not be written as a layerwise equivariant network.

with representation choice $\rho_0(h) = -1$ on the input and $\rho_2(h) = 1$ on the output. Now, we consider approximating $s$ with a neural network $f$:

$$f(x) = W_2\, \mathtt{ReLU}(W_1 x),$$

where $W_1$ and $W_2$ are both scalars. Recall (from Section 2.1) that for $\mathtt{ReLU}$ to be equivariant, the representation acting on the output of $W_1$ has to be a scaled permutation representation. The only one-dimensional permutation representation is the trivial representation, but this means that $W_1 = 0$ and so $f$ is constant 0. Also consider the non-equivariant net $\tilde{f}$ with $\tilde{W}_1 = 1, \tilde{W}_2 = 1$. Since $\tilde{f}(x) = f(x)$ for $x \leq 0$ and $\tilde{f}(x) = s(x)$ for $x > 0$, it is clear that

$$\mathbb{E}_{x \sim \mu}\|\tilde{f}(x) - s(x)\| = \frac{1}{2}\mathbb{E}_{x \sim \mu}\|f(x) - s(x)\|,$$

showing that in this example, non-equivariant nets outperform the layerwise equivariant $f$.

The reason that non-equivariant nets outperform equivariant ones in Example C.1 is that the capacity of the net is so low that requiring equivariance leads to a degenerate net. If we increase the intermediate feature dimension from 1 to 2, then equivariant nets can perfectly fit $s(x) = |x|$ (by taking $W_1 = \begin{pmatrix} 1 & -1 \end{pmatrix}^T$, $W_2 = \begin{pmatrix} 1 & 1 \end{pmatrix}$). In the next section we take this idea of insufficient capacity for equivariance a step further and give a less rigorous but more practically relevant example of a scenario where non-equivariant nets outperform equivariant nets on an equivariant task. The idea is that when features cooccur with group transformed versions of themselves, it can suffice for a network to recognize a feature in a single orientation. The network can then still be equivariant on data which has such cooccurring features, but it can be smaller than a layerwise equivariant network which would be forced to be able to recognize features in all group transformed orientations.

### C.1 Cooccurence of equivariant features

Here we give a more practical example to answer the question (C.Q) by Elesedy and Zaidi [12].

*Example* C.2. This example is illustrated in Figure 10. We consider image classification using a CNN. For simplicity we will consider invariance to horizontal flipping of the images, but the argument works equally well for other transformations. For the sake of argument, assume that our CNN has very limited capacity and can only pick up the existence of a single local shape in the input image. The CNN might consist of a single convolution layer with one filter, followed by $\mathtt{ReLU}$, and spatial average pooling. As shown in Figure 10, if the identifiable features in the images always cooccur in an image with horizontally flipped versions of themselves, then it suffices to recognise a feature in a single orientation for the feature detector to be invariant to horizontal flips *on the data*. We refer to this type of data as having *cooccurence of equivariant features*. Since a layerwise equivariant CNN with a single filter has to have a horizontally symmetric filter, it can not be as good as the non-equivariant CNN on this task.

The reader might consider Example C.2 to be somewhat artificial. However, we note that it has been demonstrated [17] that CNN image classifiers often look at mostly textures to identify classes. Textures such as the skin of an elephant will often occur in multiple orientations in any given image of an elephant. A network might hence make better use of its limited capacity by being able to identify multiple textures than by being able to identify a single texture in multiple orientations.

## D   Rewriting networks to be layerwise equivariant

The aim of this section is to investigate when we can, given an equivariant net, rewrite the net to be layerwise equivariant. I.e., given an equivariant find a functionally equivalent layerwise equivariant net with the same layer dimensions as the original. We first give a result for small two-layer networks showing that for them we can always do this. Then we look at the problem more generally in Section D.1. Unfortunately the conclusion there will not be as clear. While we can view an equivariant network as a composition of equivariant functions in a certain sense, it remains unclear whether we can rewrite it as a layerwise network with the same layer structure as the original.

**Proposition D.1.** *Consider the case of two-layer networks*

$$f(x) = W_2 \, \texttt{ReLU}(W_1 x),$$

*where* ReLU *is applied point-wise, the input dimension is* 2*, the output dimension* 1 *and the intermediate dimension* 2 *so that* $W_1 \in \mathbb{R}^{2 \times 2}$*,* $W_2 \in \mathbb{R}^{1 \times 2}$*. Any such* $f$ *that is invariant to permutations of the two input dimensions must be equivalent to a layerwise equivariant network with the same layer dimensions as* $f$*.*

*Proof.* Note that non-negative constants factor through ReLU so that

$$W_2 \begin{pmatrix} \alpha & 0 \\ 0 & \beta \end{pmatrix} \texttt{ReLU}(W_1 x) = W_2 \, \texttt{ReLU}\left( \begin{pmatrix} \alpha & 0 \\ 0 & \beta \end{pmatrix} W_1 x \right),$$

for any $\alpha \geq 0$, $\beta \geq 0$. Hence we can assume that $W_2$ only contains elements $1, -1$ by factoring the non-negative constants into $W_1$. So (WLOG) we will only have to consider $W_2 = (1 \quad \pm 1)$.

Let

$$J = \begin{pmatrix} 0 & 1 \\ 1 & 0 \end{pmatrix}$$

be the permutation matrix that acts on the input. It will also be convenient to introduce the notation $f_2(x) = W_2 \, \texttt{ReLU}(x)$ for the last part of the network so that we can write the invariance condition $\forall x \in \mathbb{R}^2 : f(Jx) = f(x)$ as $\forall x \in \mathbb{R}^2 : f_2(W_1 Jx) = f_2(W_1 x)$.

We first consider the case when $W_1$ has full rank. In this case $\forall x \in \mathbb{R}^2 : f_2(W_1 Jx) = f_2(W_1 x)$ is equivalent to

$$\forall x \in \mathbb{R}^2 : f_2(W_1 J W_1^{-1} x) = f_2(x). \tag{5}$$

For convenience we set

$$W_1 J W_1^{-1} = A = \begin{pmatrix} a & b \\ c & d \end{pmatrix}$$

which satisfies $A^2 = I$ ($A$ is an involution)[4]. It is easy to show that this means that either

$$A = \begin{pmatrix} a & b \\ c & -a \end{pmatrix},$$

where

$$a^2 + bc = 1, \tag{6}$$

or $A = \pm I$. However, the $A = \pm I$ case is easy to rule out since it implies

$$\pm W_1 = W_1 J,$$

meaning that the columns of $W_1$ are linearly dependent, but we assumed full rank. Explicitly writing out (5) with $W_1 J W_1^{-1} = A$, we get that for all $x \in \mathbb{R}^2$ we must have

$$W_2 \, \texttt{ReLU}(Ax) = W_2 \, \texttt{ReLU}(x). \tag{7}$$

Now we divide into two cases depending on the form of $W_2$. The strategy will be to plug in the columns of $A$ as $x$ and use the fact that $A^2 = I$.

---

[4]Here we note that since $(W_1 J W_1^{-1})^2 = I$, the map $\rho(J) = W_1 J W_1^{-1}$ actually defines a representation of $S_2$ on $\mathbb{R}^2$. We are hence now trying to find for which representations $\rho$ of $S_2$, $f_2$ is invariant.

- Case I: $W_2 = (1 \quad -1)$. By plugging $x = (a \quad c)^T$ into (7) we see that

$$1 = \text{ReLU}(a) - \text{ReLU}(c),$$

meaning that $a \geq 1$. Similarly, plugging in $x = (b \quad -a)^T$ we see that

$$-1 = \text{ReLU}(b) - \text{ReLU}(-a),$$

implying that $a \leq -1$. We have reached a contradiction and can conclude $W_2 \neq (1 \quad -1)$.

- Case II: $W_2 = (1 \quad 1)$. This time we will need four plug-and-plays with (7).

  1. $x = (-a \quad -c)^T$ yields

  $$0 = \text{ReLU}(-a) + \text{ReLU}(-c),$$

  so $a \geq 0$ and $c \geq 0$.
  2. $x = (-b \quad a)^T$ yields
  $$0 = \text{ReLU}(-b) + \text{ReLU}(a),$$

  so $a \leq 0$ and $b \geq 0$. From this and the previous we conclude $a = 0$.
  3. $x = (a \quad c)^T$ yields
  $$1 = \text{ReLU}(a) + \text{ReLU}(c),$$

  which implies $c \leq 1$.
  4. $x = (b \quad -a)^T$ yields
  $$1 = \text{ReLU}(b) + \text{ReLU}(-a),$$

  which implies $b \leq 1$.

  From $a = 0$ and $(b, c) \in [0, 1]^2$ we conclude from (6) that $b = c = 1$. Thus, if $W_2 = (1 \quad 1)$, then $A = J$, which means that
  $$W_1 J W_1^{-1} = J.$$
  But this is precisely the equivariance condition on $W_1$, and since $W_2$ is invariant we have a layer-wise equivariant network.

Next, we consider the case when $W_1$ has rank 1. We can then write $W_1 = uv^T$ for some vectors $u$, $v$ in $\mathbb{R}^2$. If $v$ contains two equal values, then $W_1$ is row-wise constant, so permutation invariant, and thus the network is layer-wise equivariant. If $u$ contains a zero, then a row of $W_1$ is zero and the statement to prove reduces to the case of a network with $W_1 \in \mathbb{R}^{1 \times 2}$, $W_2 \in \{-1, 1\}$ in which case it is easy to show that $W_1$ has to be permutation invariant for the network to be invariant.

Finally if $v$ contains two different values and $u$ does not contain a zero, let WLOG $v_1 \neq 0$ and consider the following two cases for the invariance equation below

$$W_2 \text{ReLU}(uv^T x) = W_2 \text{ReLU}(uv^T J x).$$

- Case I: $u$ contains two values with the same sign. We plug in $x_1 = v_1 - v_2$, $x_2 = v_2 - v_1$ and find that $uv^T x = (v_1 - v_2)^2 u$, while $uv^T J x = -(v_1 - v_2)^2 u$. One of those is mapped to 0 by ReLU and the other one isn't, implying that $W_2$ has to map it to 0, so $W_2 = (1 \quad -1)$ and the two values in $u$ are the same. This leads to a zero network which can be equivalently written layer-wise equivariant.

- Case II: $u_1$ and $u_2$ have different signs. WLOG let $u_1 > 0$. Plugging in the same values as above, we find that $W_1 = (1 \quad 1)$ and $u_2 = -u_1$. Next, we plug in $x_1 = \text{sign}(v_1)$, $x_2 = 0$ yielding $uv^T x = |v_1| u$ and $uv^T J x = \text{sign}(v_1) v_2 u$. Since $v_2 \neq v_1$ we must now have that $v_2 = -v_1$. However, $u_2 = -u_1$ and $v_2 = -v_1$ in fact yields an equivariant

$$W_1 = \begin{pmatrix} u_1 v_1 & -u_1 v_1 \\ -u_1 v_1 & u_1 v_1 \end{pmatrix}$$

and since $W_2$ is invariant, we have a layer-wise equivariant network.

$\square$

## D.1 Equivariant composition of functions

In this section we switch to a more abstract formulation of layerwise equivariance to see what it means for individual functions $\phi$ and $\psi$ if their composition $\psi \circ \phi$ is equivariant. For a related discussion of equivariance under general group actions we refer the reader to [32].

**Definition D.2.** Given a group $G$ and a set $X$, a *group action* $\alpha$ of $G$ on $X$ is a function

$$\alpha : G \times X \to X$$

such that for the identity element $i$ of $G$ and any $x \in X$ we have

$$\alpha(i, x) = x. \tag{8}$$

and for any two $g, h \in G$ and any $x \in X$ we have

$$\alpha(hg, x) = \alpha(h, \alpha(g, x)). \tag{9}$$

First of all it is practical to recall that invertible functions transfer group actions from domain to codomain and vice versa.

**Lemma D.3.** *Given a group $G$ and an invertible function $\phi : X \to Y$.*

1. *If there is a group action $\alpha_X$ on $X$, then we can define a group action $\alpha_Y$ on $Y$ by*

   $$\alpha_Y(g, y) = \phi(\alpha_X(g, \phi^{-1}(y)))$$

   *and $\phi$ is equivariant with respect to this group action. Furthermore, this is the unique group action on $Y$ that makes $\phi$ equivariant.*

2. *Similarly, if there is a group action $\alpha_Y$ on $Y$, then we can define a group action $\alpha_X$ on $X$ by*

   $$\alpha_X(g, x) = \phi^{-1}(\alpha_Y(g, \phi(x)))$$

   *and $\phi$ is equivariant with respect to this group action. Furthermore, this is the unique group action on $X$ that makes $\phi$ equivariant.*

*Proof.* We prove 1., the proof of 2. is analogous. If there is a group action $\alpha_Y$ on $Y$ making $\phi$ equivariant, then we must have

$$\alpha_Y(g, \phi(x)) = \phi(\alpha_X(g, x)),$$

for all $x \in X$. So by a change of variable $x \rightsquigarrow \phi^{-1}(t)$,

$$\alpha_Y(g, t) = \phi(\alpha_X(g, \phi^{-1}(t))),$$

for all $t \in X$. This proves uniqueness. Finally, $\alpha_Y$ is a group action as

$$\alpha_Y(i, y) = \phi(\alpha_X(i, \phi^{-1}(y))) = \phi(\phi^{-1}(y)) = y$$

and

$$\begin{aligned}
\alpha_Y(hg, y) &= \phi(\alpha_X(hg, \phi^{-1}(y))) \\
&= \phi(\alpha_X(h, \alpha_X(g, \phi^{-1}(y)))) \\
&= \phi(\alpha_X(h, \phi^{-1} \circ \phi(\alpha_X(g, \phi^{-1}(y))))) \\
&= \alpha_Y(h, \phi(\alpha_X(g, \phi^{-1}(y)))) \\
&= \alpha_Y(h, \alpha_Y(g, y)).
\end{aligned}$$

$\square$

We can now state a proposition telling us that if a composite function is equivariant, then it must be a composite of equivariant functions.

**Proposition D.4.** *Assume that we are given a group $G$ and two sets $X$ and $Z$ with group actions $\alpha_X$ and $\alpha_Z$. Assume also that we are given a further set $Y$ and two functions $\phi : X \to Y$ and $\psi : Y \to Z$ such that their composition $\psi \circ \phi : X \to Z$ is $G$-equivariant w.r.t. $\alpha_X$ and $\alpha_Z$.*

1. *If $\psi$ is invertible we can define a group action $\alpha_Y$ on $Y$ such that $\phi$ and $\psi$ are G-equivariant with respect to this action.*

2. *If $\psi$ is not invertible, introduce $Y' = \phi(X)/\sim$ where $\sim$ is the equivalence relation identifying elements of $Y$ that map to the same element of $Z$, i.e.,*

$$y_1 \sim y_2 :\Longleftrightarrow \psi(y_1) = \psi(y_2).$$

*Then we can define a group action $\alpha_{Y'}$ on $Y'$ such that $\phi$ and $\psi$ are equivariant when seen as functions to/from $Y'$.*

*Proof.* Assume first that $\psi$ is invertible. From Lemma D.3 we get a group action $\alpha_Y$ on $Y$ defined by

$$\alpha_Y(g, y) = \psi^{-1}(\alpha_Z(g, \psi(y))).$$

Lemma D.3 also states that $\psi$ is equivariant with this choice of $\alpha_Y$. Next we show that $\phi$ is equivariant, which follows from the fact that $\psi \circ \phi$ is equivariant:

$$\begin{aligned}
\alpha_Y(g, \phi(x)) &= \psi^{-1}(\alpha_Z(g, \psi \circ \phi(x))) \\
&= \psi^{-1}(\psi \circ \phi(\alpha_X(g, x))) \\
&= \phi(\alpha_X(g, x)).
\end{aligned}$$

If $\psi$ is not invertible, then $\psi$ is anyway invertible as a map $\psi : Y' \to \psi \circ \phi(X)$. Note that the action $\alpha_Z$ is well defined even when restricting from $Z$ to $\psi \circ \phi(X)$, because by the equivariance of $\psi \circ \phi$, $\alpha_Z$ can't move an element from $\psi \circ \phi(X)$ to $Z \setminus \psi \circ \phi(X)$:

$$\alpha_Z(g, \psi \circ \phi(x)) = \psi \circ \phi(\alpha_X(g, x)) \in \psi \circ \phi(X).$$

Hence the earlier argument for constructing $\alpha_Y$ works for constructing a group action $\alpha_{Y'}$ on $Y'$ such that the maps $\phi$ and $\psi$ (appropriately restricted) become equivariant. $\qquad\square$

Looking back at Example 2.1, with $\phi = W_1$, $\psi = W_2 \texttt{ReLU}(\cdot)$, we see that $Y'$ in that case would be a single point space since $W_2$ maps everything to $0$. The action $\alpha_{Y'}$ would hence be trivial.

As mentioned before, the main problem with changing $Y$ to $Y'$ as above is that if $X, Y, Z$ are feature spaces in a neural network $\psi \circ \phi$, where $\phi$ and $\psi$ themselves might be decomposable into multiple linear layers and activation functions, there is no guarantee that $Y'$ will be a vector space or that $\phi$ and $\psi$ can still be expressed as decompositions into linear layers and activation functions when changing from $Y$ to $Y'$. Still, the intuition that the equivariance should be preserved layerwise holds in the interpretation provided by Proposition D.4.

## E  Network layers with bias

In this section we generalize Proposition 2.3 to work for layers with bias, i.e. $x \mapsto Wx + t$ for some bias vector $t$. The discussion up to Section 2.1 works with biases as well, as does Lemma 2.2 (as it regards ReLU and not the layers themselves). For Proposition 2.3 it gets more complicated. What makes the proof of easy is that given an invertible linear layer $\ell(x) = Wx$, a group representation $\rho(g)$ on $x$ is transferred to a group representation $\alpha(g, x) = \ell(\rho(g)\ell^{-1}(x)) = W\rho(g)W^{-1}x$ on $\ell(x)$ w.r.t. which $W$ is equivariant. If we consider an affine layer $\ell(x) = Wx + t$ with bias vector $t$, then we can play the same game, but $\rho(g)$ is not transferred to a group representation (linear action) but instead to the affine action $\alpha(g, x) = W\rho(g)W^{-1}x + (I - W\rho(g)W^{-1})t$ on $\ell(x)$. This means that we can not apply Lemma 2.2. We can however find a generalization as follows:

**Lemma E.1.** *If $A$, $B$ are $n \times n$ invertible matrices and $a$ and $b$ are $n$-vectors, such that $\texttt{ReLU}(Ax + a) = B\texttt{ReLU}(x) + b$ for all $x \in \mathbb{R}$, then $a = b = 0$.*

*Proof.* Inserting $x = 0$ yields $\texttt{ReLU}(a) = b$. Inserting $x = A^{-1}a$ yields $\texttt{ReLU}(2a) = B\texttt{ReLU}(A^{-1}a) + b$ so that $b = B\texttt{ReLU}(A^{-1}a)$. Inserting $x = -A^{-1}a$ yields $0 = B\texttt{ReLU}(-A^{-1}a) + b$ so $b = -B\texttt{ReLU}(-A^{-1}a)$. Combined we have that $B^{-1}b = \texttt{ReLU}(A^{-1}a) = -\texttt{ReLU}(-A^{-1}a)$ so that $B^{-1}b$ must be zero and hence so must $b$. Finally, inserting $x = -2A^{-1}a$ yields $\texttt{ReLU}(-a) = B\texttt{ReLU}(-2A^{-1}a) + b$ so that $\texttt{ReLU}(-a) = -b = 0$ which combined with $\texttt{ReLU}(a) = b = 0$ gives $a = 0$. $\qquad\square$

This lemma shows that if `ReLU` commutes with affine actions, then the affine actions must in fact be linear and so Lemma 2.2 applies. This shows that Proposition 2.3 holds with affine layers as well. We thank the anonymous reviewer who prompted this generalization.

