# OpenReview forum: "Investigating how ReLU-networks encode symmetries"
_NeurIPS.cc/2023/Conference — NeurIPS 2023 poster_

### Official Review · Reviewer_ZqAK · 2023-06-19

**Soundness:** 3 good
**Presentation:** 3 good
**Contribution:** 3 good
**Rating:** 7
**Confidence:** 3

**Summary:**

The paper considers networks with ReLU activations and investigates in which specific way their internal activations learn to be equivariant given an invariant data distribution.
Section 2 presents a theoretical result, applying specifically to two-layer networks with non-singular weight matrices. It is shown that the network's equivariance requires 1) that its input and output activations are acted on by scaled permutation representations, and 2) that the network is layerwise equivariant with a related scaled permutation representation acting on the hidden features. Example 2.1 clarifies that this result does not necessarily hold in the case of non-singular weights.
The third section proposes the conjecture that ReLU-CNNs that are trained on a reflection-invariant data distribution are close to regular GCNNs, i.e. GCNNs whose group representations are permutation representations. This conjecture is based on Entezari et al.'s conjecture that the activations of neural networks can always be permuted such that the networks are closely connected in the loss landscape. The authors argue that there always exists a CNN with exactly flipped kernels at initialization, which is due to the assumed invariant distribution preserved throughout training. Since these nets can be Entezari's conjecture be aligned via channel permutations, it follows that the CNN has either invariant kernels or kernels which are related by reflections - they are hence layerwise equivariant regular GCNNs.
The experimental section investigates whether these two results hold in practice, using an activation matching technique to find reflected kernels. The results suggest that layerwise equivariance holds indeed.

**Strengths:**

The paper is well written and gives insight into the nature of networks which are trained to be equivariant. This is an original idea since most publications on equivariant networks are rather investigating models which are by design constrained to be equivariant. The actual necessity of regular group convolutions is clarified by the submission.
Where possible, the networks layerwise equivariance is undoubtably proven with an analytical approach. While this approach does not generalize to networks with more than two layers or singular weight matrices, the authors use a different approach to tackle these cases. All results and conjectures are supported empirically.
The extent of the paper is comprehensive, containing multiple additions in the supplementary material.

**Weaknesses:**

The strong assumptions in the analytical result seem initially like a weakness, but are addressed by the alternative approach based on Entezari's conjecture. The argument that this conjecture implies layerwise equivariance seems reasonable, but could be explained better. I am also not sure about the conjecture applies in the first place, however, this is not the scope of the current paper.
Technically, I do not see the necessity for the definition of the barrier in Eq. (4) exact at the parameter's average, instead of at the maximizing parameter value.
Another weakness is that the authors are only considering reflection-equivariant GCNNs, but not more general symmetry groups.

**Questions:**

Why is section 2 considering linear instead of affine layers? Do the results not hold when biases are summed?
Lemma 2.2 claims that the diagonal scaling matrix would be required to have (strictly) positive entries, however, non-negative entries, including zeros, seem sufficient. The resulting set of matrices does then no longer form a group since invertibility is lost, but the original group is contained as subset. The "intertwiner group" should furthermore be renamed to "equivariance group" since intertwiners are by definition linear, which ReLU is not.
The activation matching technique should be briefly explained in the current submission instead of only pointing to related work.
The paper is currently only mentioning trivial and regular representations, however, everything should apply to more general quotient representations as well, which should be briefly clarified.

**Limitations:**

Limitations are clearly addressed in section 1.2. and societal impacts are not to be expected.

---

> ### Author Rebuttal · Authors · 2023-08-07
>
> We thank the reviewer for their valuable feedback and helpful review.
>
> > Why is section 2 considering linear instead of affine layers? Do the results not hold when biases are summed?
>
> This is an excellent question. We wrote it this way (1) for simplicity, (2) because Elesedy & Zaidi [9] use the same setting and (3) for the fact that Prop 2.3 deals with bias free layers. The discussion up to Section 2.1 works with biases as well, as does the Lemma 2.2 (as it regards ReLU and not the layers themselves). The paragraph after Lemma 2.2 also works fine. For Proposition 2.3 it gets more complicated. What makes the proof of 2.3 easy is that given an invertible linear layer $\ell:X\to Y$, $\ell(x)= Wx$, a group representation $\rho(g)$ on $X$ is transferred to a group representation $\alpha(g, y) =\ell(\rho(g)\ell^{-1}(y))=W\rho(g)W^{-1}y$ on $Y$ w.r.t. which $\ell$ is equivariant. If we consider an affine layer $\ell(x) = Wx + t$ with bias vector $t$, then we can play the same game, but $\rho(g)$ is not transferred to a group representation (linear action) but instead to the affine action $\alpha(g, y) = \ell(\rho(g)\ell^{-1}(y))= W\rho(g)W^{-1}y  + (I - W\rho(g)W^{-1})t$ on $Y$. This means that we can not apply Lemma 2.2.
> We can however find a generalization as follows:
>
> **Lemma** If $A$, $B$ are $n\times n$ invertible matrices and $a$ and $b$ are $n$-vectors, such that $\mathtt{ReLU}(Ax + a) = B\mathtt{ReLU}(x) + b$ for all $x\in\mathbb{R}^n$, then $a=b=0$.
>
> *Proof:* Inserting $x=0$ yields $\mathtt{ReLU}(a)=b$. Inserting $x=A^{-1}a$ yields $\mathtt{ReLU}(2a)=B\mathtt{ReLU}(A^{-1}a)+b$ so that $b=B\mathtt{ReLU}(A^{-1}a)$. Inserting $x=-A^{-1}a$ yields $0=B\mathtt{ReLU}(-A^{-1} a) + b$ so $b=-B\mathtt{ReLU}(-A^{-1}a)$. Combined we have that $B^{-1}b=\mathtt{ReLU}(A^{-1}a)=-\mathtt{ReLU}(-A^{-1}a)$ so that $B^{-1}b$ must be zero and hence so must $b$. Finally, inserting $x=-2A^{-1}a$ yields $\mathtt{ReLU}(-a)=B\mathtt{ReLU}(-2A^{-1}a) + b$ so that $\mathtt{ReLU}(-a)=-b=0$ which combined with $\mathtt{ReLU}(a)=b=0$ gives $a=0$.
>
> This lemma shows that if ReLU commutes with affine actions, then the affine actions must in fact be linear and so Lemma 2.2 applies. This shows that Proposition 2.3 holds with affine layers as well. We would like to greatly thank the reviewer for prompting this generalization which we will include in the appendix.
>
> > Lemma 2.2 claims that the diagonal scaling matrix would be required to have (strictly) positive entries, however, non-negative entries, including zeros, seem sufficient. The resulting set of matrices does then no longer form a group since invertibility is lost, but the original group is contained as subset.
>
> It is correct that non-negative diagonal matrices commute with ReLU. The reason for excluding them in the lemma is that otherwise we would have to exclude them later when talking about group representations, for the reason the reviewer states - non-invertibility.
>
> > The "intertwiner group" should furthermore be renamed to "equivariance group" since intertwiners are by definition linear, which ReLU is not.
>
> This terminology comes from Godfrey et al. 2022. We agree with the reviewer that the naming is a bit unfortunate. We are unsure if the best course of action would be to add a footnote or to simply remove the sentence, and are happy to receive further feedback.
>
> >  The activation matching technique should be briefly explained in the current submission instead of only pointing to related work. The paper is currently only mentioning trivial and regular representations, however, everything should apply to more general quotient representations as well, which should be briefly clarified.
>
> We thank the reviewer for these helpful suggestions. We would like to clarify that trivial and regular representations are the only ones discussed for the horizontal flipping case as any permutation representation of the 2 element cyclic group must consist of fixed elements and transpositions only. I.e. the permutation representations are direct sums of trivial and regular representations.

---

> > ### Comment · Reviewer_ZqAK · 2023-08-11
> >
> > Thanks for the detailed response. The rebuttal addresses all of my questions and I would like to express again that I vote for accepting the paper.
> >
> > Regarding the "intertwiner group" terminology I think that both of the proposed solutions are fine. I would personally choose "equivariance group" as it should be understood by everyone working on equivariant networks and does not require the additional footnote. Note that not every reader in equivariant DL may be familiar with representation theory, intertwiners or the terminology of Godfrey et al.

---

### Official Review · Reviewer_79cV · 2023-07-07

**Soundness:** 4 excellent
**Presentation:** 3 good
**Contribution:** 3 good
**Rating:** 7
**Confidence:** 3

**Summary:**

This work investigates the relationship between end-to-end equivariance of a network and layerwise equivariance. It theoretically investigates when we can guarantee that an equivariant network is layerwise equivariant, and also cases where layerwise equivariant is not guaranteed or is harmful. In the case of CNNs, the authors draw a connection between horizontal flip invariance and the permutation conjecture for linear mode connectivity.

**Strengths:**

1. Good exposition in Sections 1 and 2
2. Appendices C and D have nice observations and are illustrative. I quite like the examples in C and C.1., which concretely show ways in which layerwise equivariance is not good enough when network capacity is low.
3. Neat simpler proof of the lemma from Godfrey et al. 2022
4. Nice connection drawn to the permutation conjecture that I would not have expected.
5. The trained VGGs are remarkably similar to GCNNs.

**Weaknesses:**

1. The point about layerwise equivariance is covered in some prior works that are not cited. Much of appendix D.1. in particular is discussed in depth in [1]. Limitations of linear layerwise equivariance is discussed in [Finzi et al. 2021].
2. As noted by the authors, the restriction to horizontal flips is restrictive for empirical evaluation, though to be fair horizontal flips consistently improve many vision systems.

[Shakerinava et al. 2022] Structuring Representations Using Group Invariants. NeurIPS 2022.
[Finzi et al. 2021] A Practical Method for Constructing Equivariant Multilayer Perceptrons for Arbitrary Matrix Groups. ICML 2021


**Questions:**

What do you mean on Page 5 when you say that the results for Godfrey et al. 2022 for other nonlinearities "also straightforwardly translate to the group equivariance case"?

Minor:
* Page 4: "and then $\tilde f$ is equivariant" should be "... layerwise equivariant"
* Can you give a citation for the existence of dead neurons on page 4?
* Page 6, several times you refer to the numerator of (4) when you mean the denominator.


**Limitations:**

Good discussion of limitations in Sectino 1.2.

---

> ### Author Rebuttal · Authors · 2023-08-07
>
> We thank the reviewer for their useful suggestions and helpful review.
>
> > The point about layerwise equivariance is covered in some prior works that are not cited. Much of appendix D.1. in particular is discussed in depth in [1]. Limitations of linear layerwise equivariance is discussed in [Finzi et al. 2021].
>
> Thank you for pointing out these overlooked references. We will of course incorporate them.
>
> > What do you mean on Page 5 when you say that the results for Godfrey et al. 2022 for other nonlinearities "also straightforwardly translate to the group equivariance case"?
>
> We apologize for the sloppy formulation which we will improve. What is meant is that Section 3.1/Table 1 of Godfrey et al. contains several results of the type (paraphrased) "For nonlinearity $\sigma$ if $\sigma(Ax)=B\sigma(x)$ for invertible matrices $A, B$ then $A$ and $B$ have the following specific forms...". This implies that the representations acting on the feature spaces in a layerwise equivariant network with activation function $\sigma$ must have matrices of the respective forms of $A$ and $B$.
>
> > Minor [...]
>
> Thanks!

---

> > ### Comment · Reviewer_79cV · 2023-08-11
> >
> > Thank you for the response! I have no further questions at the time, and will discuss with other reviewers.

---

### Official Review · Reviewer_k3h9 · 2023-07-07

**Soundness:** 4 excellent
**Presentation:** 2 fair
**Contribution:** 4 excellent
**Rating:** 8
**Confidence:** 3

**Summary:**

This paper provides an investigation on whether equivariance of a trained deep neural network with ReLU activations implies that each of its learned layers are equivariant. The authors show that this should be true in some sense, i.e., some kind of group action must be present in the intermediate feature spaces (Line 141-143 and Appendix D.1), as the network should be able to always encode the group transformation of the input in some way to achieve overall equivariance. However, this applies to intermediate feature spaces where the clamping behavior of (ReLU) activations are involved; the authors argue that, when considering the learned linear layers, this is not always true (Example 2.1). Regarding ReLU activation, the authors show that if ReLU is G equivariant, the representations on the input and output must be scaled permutation representations (Lemma 2.2). From that, for two-layer networks with a strong assumption of invertible weight matrices, the authors show that overall equivariance implies layerwise equivariance where representations on intermediate features are scaled permutation representations (Proposition 2.3). Based on that, the authors investigate how the (requirement for) scaled permutation representations provides an implication of how horizontal flipping symmetry (related to the representation of S2 group) can be encoded in CNNs. The key intuition here is that permutation representation exactly appears in the characterization of GCNNs (Cohen et al., 2016), where the kernel constraint gives that an element of S2 group acts on the group convolution kernel through the joint action of spatial permutation representation (horizontal flipping here) and channel permutation representation (allocation). The authors further make a connection to the invariance of neural networks under permutation of neurons, or channels in case of CNNs due to spatial structure of convolutions, and the permutation conjecture (Entezari et al., 2022) that networks of the same type trained on the same data would lie in the same loss basin up to some permutation of neurons (channels). Then, the authors logically combine the requirement for permutation representations on ReLU for G equivariance, the spatial and channel permutation representation on group convolution filters, and the permutation conjecture, leading to the following conjecture: CNNs trained on a data distribution invariant to horizontal flips would be close to being GCNNs. The authors empirically test their conjecture using VGG and ResNet architectures on CIFAR10 and ImageNet, which supports the proposed conjecture.

Cohen et al., Group Equivariant Convolutional Networks (2016)

Entezari et al., The role of permutation invariance in linear mode connectivity of neural networks (2022)

**Strengths:**

S1. I think this is a solid work that establishes a creative combination of ideas and findings from multiple sub-areas of deep learning theory, and from that proposes some original theoretical findings that combines into a very interesting conjecture that is potentially impactful in the field of equivariant deep learning.

S2. In addition to the above, the proposed conjecture is equipped with a proper empirical support involving multiple deep architectures and datasets, which is an important strength of this work.

**Weaknesses:**

W1. While reading the paper, I was quite confused about the implication of formulation in Appendix D.1. It seems the results, in particular Proposition D.4, explicitly proves that equivariance of a network implies layerwise equivariance, even under non-invertibility of individual layers. Because of this, I was confused when reading the discussion on Example 2.1 as well as Line 158 that overall equivariance might not lead to layerwise equivariance, as these seem like conflicting arguments. Am I missing something?

W2. The discussion on identifying an equivalent equivariant network with layerwise equivariance (Line 132-136), while understandably leading to discussion in Appendix C, seems not critical in describing the main conjecture of the paper (please correct me if I am wrong). It might be better in terms of readability to contain the relevant discussion in some separated section.

W3. In describing Eq. (3), it might be beneficial to explain how the transformation of filter relates to transformation of input and output (in context of equivariance) for readers not familiar with GCNN.

W4. The description of why and how the REPAIR algorithm is used in Line 289-296 was hard to understand, I think it has room for improvement.

**Questions:**

Q1. Reading Line 260-268, a question I had was to which extent the proposed conjecture depends on the particular nature of S2, i.e., how specifically it may generalize to other discrete groups such as 90-degree rotations. May I ask for an explanation on this?

**Limitations:**

The authors have clarified the limitation of the work in Section 1.2.

---

> ### Author Rebuttal · Authors · 2023-08-07
>
> We thank the reviewer for their review and helpful comments.
>
> > While reading the paper, I was quite confused about the implication of formulation in Appendix D.1. It seems the results, in particular Proposition D.4, explicitly proves that equivariance of a network implies layerwise equivariance, even under non-invertibility of individual layers. Because of this, I was confused when reading the discussion on Example 2.1 as well as Line 158 that overall equivariance might not lead to layerwise equivariance, as these seem like conflicting arguments. Am I missing something?
>
> This is an excellent questions and something we have struggled with ourselves. The way to interpret it is that as linear functions between vector spaces, the layers need not be equivariant. But if we redefine the domains/codomains of the layers then they become equivariant functions with respect to specific nonlinear group actions. In particular, the framing in Appendix D does not rule out group actions in the middle of the network which are nonlinear. In the main part of the paper we discuss the case of linear group actions (=representations) acting on all feature spaces.
>
> > W2, W3, W4
>
> We thank the reviewer for these helpful suggestions.
>
> > Reading Line 260-268, a question I had was to which extent the proposed conjecture depends on the particular nature of S2, i.e., how specifically it may generalize to other discrete groups such as 90-degree rotations. May I ask for an explanation on this?
>
> In the 90-degree rotation case, Entezari's conjecture plus the assumption that the data distribution is rotation invariant gives that when we rotate all filters of the initial CNN 90 degrees, there should be a permutation that aligns these rotated filters with the original ones. If this permutation is cyclic of order 4, then it is a permutation representation of the group of 90 degree rotations. Thus the CNN is a G-CNN w.r.t. this group.

---

> > ### Comment · Reviewer_k3h9 · 2023-08-18
> > **Response to rebuttal**
> >
> > Thank you for the detailed response. I have read through the manuscript again with the response in mind, and the narrative leading to ReLU and permutation representation is more clear now. I recommend the authors to revise Line 141-148 according to the rebuttal to prevent potential misunderstanding (like the one I had) regarding Proposition D.4. Now that my major concern is resolved, I have adjusted my score accordingly.

---

### Official Review · Reviewer_A92L · 2023-07-08

**Soundness:** 3 good
**Presentation:** 4 excellent
**Contribution:** 3 good
**Rating:** 6
**Confidence:** 3

**Summary:**

Exploring the symmetries of representations and parameters in neural networks is crucial. This paper provides several valuable contributions. First, the authors theoretically proved that for Relu-Networks equivariance implies layer-wise equivariance, but not vice versa. Second, inspired from the conjecture by [22], the authors discussed a weaker version that connects CNNs with GCNNs. Finally, quantitative experiments were performed and showed that a ReLU-network that has been trained to be equivariant will be layer-wise equivariant.

**Strengths:**

1.	This paper is well written. I enjoy the writing. While the concepts particularly those related to group representation are complicated and non-trivial, the authors did a great job in introducing the relevant definitions, examples and propositions in an elegant and compact way. The authors also provided necessary intuitive understanding, which greatly help readers to digest the motivation of how the investigation is performed.

2.	The conclusion by Proposition 2.3 is interesting and valuable, even the proof is a direct result of Lemma 2.2 by [14]. I have checked [14] and find no similar conclusion. The authors also provided abundant theoretical understandings and derivations in the supplementary materials including the discussion of the invariant case.

3.	The discussion of how GCNNs are related with the notion of Conjecture 3.1 [14] is insightful and inspiring.

4.	The experimental evaluations are scrupulously carried out and are able to support the claims by the authors to a certain extent.


**Weaknesses:**

I have no major concern. There are still some questions below:

1.	Proposition 2.3 shows that layer-wise equivariance is NOT a necessary condition of equivariance. But the experiments did show that CNN with equivariant training boils down to be layer-wise equivariant. So, why we have this observation and does it mean that the examples derived from Proposition 2.3 are just corner cases that rarely happen in practice? Moreover, how does Proposition 2.3 influence the design of GCNNs? Or particularly, what is the relationship between the equivariant constraint in Proposition 2.3 and the derived kernel constraint (Eq. 3) by [31,30]? I would expect the authors to explain more on these points?

2.	Would the authors explain why they have the claims in Lines 151-155 that equivariance can hurt the performance? Why in Table 2, the unconstrained models obtained better performance than those constrained or equivariant ones?

3.	Table 1 only defined the models trained with invariant losses, which indicates that the performance in Table 2 can only answer Q1? How about the experiments of models trained with horizontal flipping data for Q2? Is Table 3 mainly for Q2?


**Questions:**

See the weakness part above.

**Limitations:**

The authors have adequately discussed the limitations.

---

> ### Author Rebuttal · Authors · 2023-08-07
>
> We thank the reviewer for their good comments and questions which will help us improve the paper.
>
> > Proposition 2.3 shows that layer-wise equivariance is NOT a necessary condition of equivariance. But the experiments did show that CNN with equivariant training boils down to be layer-wise equivariant. So, why we have this observation and does it mean that the examples derived from Proposition 2.3 are just corner cases that rarely happen in practice?
>
> Let us clarify that Proposition 2.3 shows for simple networks that equivariance and layer-wise equivariance are equivalent. Example 2.1 however indeed shows that when the conditions in Proposition 2.3 (specifically, invertibility) do not hold, then a network may be equivariant without being layerwise equivariant. The fact that we don't observe the degeneracy from 2.1 in the experiments could be explained as follows. Note that if several input channels are set to zero by a weight matrix + ReLU, then their values do not matter for the network output. Thus even if they are not equivariant, we can permute them without changing the output of the network. So the permutation found by the weight matching procedure in the experiments might not be actually permuting these, later killed off, channels in an equivariant matter, meaning that the network is not strictly layerwise equivariant. But it is layerwise equivariant for the channels that matter.
>
> > Moreover, how does Proposition 2.3 influence the design of GCNNs? Or particularly, what is the relationship between the equivariant constraint in Proposition 2.3 and the derived kernel constraint (Eq. 3) by [31,30]? I would expect the authors to explain more on these points?
>
> This is an excellent question. While the kernel constraint says given certain group representations what the possible linear layers are, our results instead concern given a certain activation function which group representations are possible. So the results are complementary.
>
> > Would the authors explain why they have the claims in Lines 151-155 that equivariance can hurt the performance?
>
> This refers to the discussion in Appendix C, where we give specific examples. The main point is that an invariant/equivariant network must be equally good at detecting every pattern in all orientations. In practice given a limited network capacity, it may give better performance to be able to recognize more patterns in fewer orientations than fewer patterns in more orientations.
>
> > Why in Table 2, the unconstrained models obtained better performance than those constrained or equivariant ones?
>
> We do not know. However a possibility is that it relates to the mentioned discussion in Appendix C, i.e. that the invariant/equivariant networks do not have enough capacity to learn all patterns necessary to classify the data well.
>
> > Table 1 only defined the models trained with invariant losses, which indicates that the performance in Table 2 can only answer Q1? How about the experiments of models trained with horizontal flipping data for Q2? Is Table 3 mainly for Q2?
>
> All the networks in Table 1 in fact are trained with horizontal flipping data augmentation (see line 282). We will make this more explicit. We have now additionally performed experiments without flip augmentation, please refer to the global rebuttal.

---

> > ### Comment · Reviewer_A92L · 2023-08-11
> > **Thank your for the reply**
> >
> > I thank the authors for their efforst in addressing my concerns. There are some misunderstanding I have made before. But after checking the feedbacks and other reviewers' comments. I am sure that this is a solid paper and comes valuable for the publication in NeurIPS. I have no any question now.

---

### Official Review · Reviewer_k5hy · 2023-07-08

**Soundness:** 3 good
**Presentation:** 3 good
**Contribution:** 2 fair
**Rating:** 5
**Confidence:** 2

**Summary:**

This paper shows that CNNs will be close to G-CNNs if they are trained to be equivariant. In addition, they also provide some theoretical analysis and conjectures regarding the layerwise equivariant of ReLU-networks.

**Strengths:**

1. They show that equivariance implies layerwise equivariance with a scaled permutation representation acting on the feature maps (Proposition 2.3.).
2. They propose a new conjecture 'Most SGD CNN-solutions on image data with a distribution that is invariant to horizontal flips of the images will be close to GCNNs.' (Conjecture 3.2.) They proposed a new measure for closeness to being a GCNN.
3. Experiments on CIFAR-10 and ImageNet support Conjecture 3.2.

**Weaknesses:**

1. In general, the paper may have limited or vague benefits for applications. First, the work focuses on ReLu-networks in the theoretical analysis, ignoring normalization layers. Modern neural network designs heavily rely on different normalization layers. In addition, in the ImageNet experiments, ResNet-50 uses batch normalization, which is not aligned with the theoretical analysis.
2. The paper did not explain what are the insights for network designs if ReLu CNNs are layerwise equivariance. In practice, it even hurts the performance of the model if the invariance loss is applied. Then, the generalization of the model might contradict the actual layerwise equivariance of the model. And the invariance loss adds an extra restriction for learning.
3. The evidence of Q1 is much weaker than Q2 in section 4. The  G-CNN Barrier of 'CNN' in table 2 is 20$\times$ larger than 'CNN + inv-loss.' I wonder if it is still valid to say it is close to G-CNN. Maybe a baseline without horizontal data augmentation can better justify this.

**Questions:**

1. What is 'ppt' defined in section 4.2? I did not find explanations regarding it.
2. Could you give more explanations regarding the last sentence of the conclusion 'If a negative GCNN barrier is achievable ...... enable weight space ensembling.'? How 'negative GCNN barrier' is related to 'weight space test time data augmentation', etc?

**Limitations:**

Limitations are given.

---

> ### Author Rebuttal · Authors · 2023-08-07
>
> We thank the reviewer for the helpful review. We aim to clarify the writing in the paper in accordance with our replies to the reviewers comments below.
>
> > In general, the paper may have limited or vague benefits for applications.
>
> It is true that there might not be immediate benefits, but this is the case for much theoretical work on neural networks. See also the clarification of "weight space test time data augmentation" further down.
>
> > First, the work focuses on ReLu-networks in the theoretical analysis, ignoring normalization layers. Modern neural network designs heavily rely on different normalization layers. In addition, in the ImageNet experiments, ResNet-50 uses batch normalization, which is not aligned with the theoretical analysis.
>
> Batch normalization is not included in the theory section, however as the reviewer notes the experiments suggest that layerwise equivariance is still compatible with batch normalization. Note that our result says that using ReLU implies that if we have layerwise equivariance then the representations acting on the features are permutation representations. Batch normalization is equivariant w.r.t. such representations if the batch statistics are computed jointly for channels in the same cycle in the cyclic decomposition of the representation [1]. The fact that we have positive experimental results indicates that statistics for such channels are approximately the same.
>
> > In practice, it even hurts the performance of the model if the invariance loss is applied. Then, the generalization of the model might contradict the actual layerwise equivariance of the model. And the invariance loss adds an extra restriction for learning.
>
> The invariance loss is not added to boost performance, but rather to test the theory of whether an invariant CNN is layerwise equivariant. Surprisingly, the setting with invariance loss added after 20% of the training epochs did improve performance of simple VGG nets. This is a result that we have not seen in prior literature.
>
> > The evidence of Q1 is much weaker than Q2 in section 4. The G-CNN Barrier of 'CNN' in table 2 is 20 larger than 'CNN + inv-loss.' I wonder if it is still valid to say it is close to G-CNN.  Maybe a baseline without horizontal data augmentation can better justify this.
>
> The closeness to a G-CNN should be interpreted in relation to Entezari's conjecture (3.1). Our experiments suggest that a CNN trained with data augmentation is closer to being a G-CNN than it is to another independently trained CNN.
> It is indeed the case that it is not super close to being a G-CNN, but it also is not invariant to horizontal flips so this does not contradict our theory.
> In any case, a baseline w/o horizontal flipping augmentation is a great suggestion and we have carried it out. Please refer to the global rebuttal.
>
> > What is 'ppt' defined in section 4.2? I did not find explanations regarding it.
>
> It is an abbreviation for percentage point. We will write it out in full to avoid confusion.
>
> > Could you give more explanations regarding the last sentence of the conclusion 'If a negative GCNN barrier is achievable ...... enable weight space ensembling.'? How 'negative GCNN barrier' is related to 'weight space test time data augmentation', etc?
>
> One of the motivations for studying the Entezari conjecture has been that if we can find permutations relating two networks, then we can average their weights so that we obtain a so called weight space ensemble of them. The goal being to obtain the benefits of ensembling while still working with a single network. If we can find permutations relating a network A to its horizontally flipped self B, then we can perform such weight space ensembling between A and B. Note that the output of B will be the same (up to border/stride effects) as what we would get if we insert a horizontally flipped image into A. Thus ensembling A and B is the same as doing test time augmentation with horizontal flipping, and weight space ensembling of A and B could be called weight space test time augmentation. If the GCNN barrier is negative it means that we get improved performance by doing this. Note that a negative barrier has not been obtained for the Entezari setting of two different networks yet and our results suggest that it should be easier to obtain for our setting of a network and its flipped self.
>
>
>
> [1] Weiler & Cesa, General E(2)-Equivariant Steerable CNNs, NeurIPS, 2019

---

> > ### Comment · Reviewer_k5hy · 2023-08-16
> >
> > I want to thank the authors for their detailed responses. Most of my concerns are addressed and additional explanations are provided.  The 'weight space test time data augmentation' may provide an interesting direction for future works, and it seems promising. As a result, I changed my score to positive.

---

### Author Rebuttal · Authors · 2023-08-07

We thank all reviewers for their well-written and very helpful reviews.

Reviewer k5hy suggested an extra experiment which we have carried out and believe will be of interest to all reviewers.
We train VGG-11 nets on CIFAR-10 *without* horizontal flipping data augmentation. These models have lower accuracy and lower invariance than the ones trained with flip augmentation. Most importantly, they have higher GCNN barrier. The GCNN barrier for the nets trained without flip augmentation is approximately as large as the barrier between two separate nets a la Entezari et al. Since the data still contains horizontal flipping symmetries we find it non-surprising that the obtained nets are still quite close to being GCNNs in this sense. Please find the relevant numbers in the table below:

| Name                                   | Accuracy                     | Invariance Error             | Barrier                                  |
|----------------------------------------|------------------------------|------------------------------|------------------------------------------|
| CNN (Table 2)                          | $0.901 \pm 2.1\cdot 10^{-3}$ | $0.282 \pm 1.8\cdot 10^{-2}$ | $4.00\cdot 10^{-2}\pm 4.9\cdot 10^{-3}$  |
| CNN w/o horizontal flip augmentation (NEW)                | $0.879 \pm 1.8\cdot 10^{-3}$ | $0.410 \pm 4.0\cdot 10^{-2}$ | $4.98\cdot 10^{-2}\pm 6.1\cdot 10^{-3}$  |
| CNN merged with separate net (Table 4, appendix) | $0.901 \pm 2.1\cdot 10^{-3}$ | $0.282 \pm 1.8\cdot 10^{-2}$ | $5.08\cdot 10^{-2}\pm 5.7\cdot 10^{-3}$  |

Also of general interest could be that reviewer ZqAK prompted a generalization of Lemma 2.2 and Proposition 2.3 to affine layers. We refer to the individual rebuttal for this - it is the first question answered there.

---

> ### Author Response · Authors · 2023-08-22
> **Post rebuttal**
>
> We wish to thank the reviewers for going through our rebuttals and are of course very happy to read that their questions have been answered and that all reviewers recommend acceptance of the paper.

---

### Decision · Program_Chairs · 2023-09-21

**Decision:**

Accept (poster)

**Comment:**

A paper investigating whether equivariance of a network implies layerwise equivariance. For two-layer networks, a simple characterization is provided. For multi-layer CNNs, a range of numerical experiments supporting the considered thesis under horizontal flipping symmetry is given.

The reviewers found the research problem put forward in the paper novel and intriguing, and coincided that sufficient empirical evidence has been given to make the numerical results sufficiently solid. Please make sure to address critical comments raised by the reviewers; in particular, applicability beyond the simple settings studied in the work.